# Truncating tau reveals different pathophysiological actions of oligomers in single neurons

Emily Hill [1✉], Thomas K. Karikari [2], Juan Lantero-Rodriguez [2], Henrik Zetterberg [2,3,4,5], Kaj Blennow [2,3], Magnus J. Richardson[6] & Mark J. Wall [1✉]

Tau protein is involved in maintaining neuronal structure. In Alzheimer's disease, small numbers of tau molecules can aggregate to form oligomers. However, how these oligomers produce changes in neuronal function remains unclear. Previously, oligomers made from full-length human tau were found to have multiple effects on neuronal properties. Here we have cut the tau molecule into two parts: the first 123 amino acids and the remaining 124-441 amino acids. These truncated tau molecules had specific effects on neuronal properties, allowing us to assign the actions of full-length tau to different regions of the molecule. We identified one key target for the effects of tau, the voltage gated sodium channel, which could account for the effects of tau on the action potential. By truncating the tau molecule, we have probed the mechanisms that underlie tau dysfunction, and this increased understanding of tau's pathological actions will build towards developing future tau-targeting therapies.

[1] School of Life Sciences, University of Warwick, Coventry CV4 7AL, UK. [2] Department of Psychiatry and Neurochemistry, Institute of Neuroscience and Physiology, University of Gothenburg, SE-43180 Mölndal, Sweden. [3] Clinical Neurochemistry Laboratory, Sahlgrenska University Hospital, SE-43180 Mölndal, Sweden. [4] UK Dementia Research Institute at UCL, London WC1E 6BT, UK. [5] Department of Neurodegenerative Disease, UCL Institute of Neurology, London WC1E 6BT, UK. [6] Institute of Mathematics, University of Warwick, Coventry CV4 7AL, UK. ✉email: E.hill.4@warwick.ac.uk; Mark.wall@warwick.ac.uk

Tau is a microtubule-associated protein that can modify neuronal morphology, vesicle transport and trafficking[1,2]. Tau dysfunction contributes to diseases, termed tauopathies, either as the primary causative agent (e.g., Pick's disease) or as a component of the neuropathology (e.g., Alzheimer's disease, AD). Tauopathies have varying histopathological and clinical presentations and are often distinguished by the ultra-structure of the tau aggregates[3,4]. In AD, abnormally phosphorylated tau dissociates from microtubules and aggregates to form oligomers and fibrils, accumulating in the soma-dendritic compartment[5]. Tau can further aggregate to form neurofibrillary tangles (NFTs), whose abundance correlates with disease progression[6,7]. However, it appears that the soluble tau oligomers (oTau) are the most bioactive, in terms of disrupting neuronal function. Indeed, the toxic effects of tau oligomers can occur in the absence of NFT pathology[8–10]. Although the toxicity of oTau is now well-established, the mechanistic basis of oTau actions on neuronal function is poorly understood.

Electrophysiological studies have revealed several pathological mechanisms for the actions of full-length (FL) oTau, including alterations in neuronal excitability and short- and long-term synaptic plasticity[11–14]. Viral introduction and transgenic models are commonly used to explore the roles of oTau in pathology[8–10]. While these methods generate valuable data, they provide little information on the concentration and structural conformations of oTau that are responsible for the neurotoxic effects. For other approaches, such as the extracellular application of oTau to tissue or cell cultures[15–17], cellular uptake may be the limiting step, reducing the observed toxicity. Moreover, it is not possible to target oligomers to either pre- or post-synaptic neurons and homeostatic adaptation may occur.

To address these limitations, whole-cell patch-clamp recording has recently been used to introduce oTau into single neurons and has enabled specific targeting to either pre- or post-synaptic cells[11,18]. This approach only requires small amounts of oligomers, each cell acts as its own control and electrophysiological alterations can be observed in real time. Furthermore, direct comparisons can be made between different concentrations and structural conformations of the oTau introduced[11,19,20].

Previously, full-length (FL, prepared from full-length human tau 1–441) oTau (44–444 nM) was shown to modify the excitability (as measured by a change in firing rate), input resistance, and action-potential waveform of CA1 pyramidal neurons[11]. We hypothesised that specific truncations of the tau molecule may allow these various effects to be dissected apart, identifying the underlying mechanisms and which parts of the molecule cause which effect. We therefore created truncated versions of tau informed by a physiologically relevant truncation at amino acid 123 that was recently discovered in AD patients[21]. We used oligomers formed from tau lacking the N-terminal region (aa 124–444; C$^{FRAG}$) and soluble aggregates of the non-oligomer forming N-terminal fragment (aa 1–123; N$^{FRAG}$). Using these tau constructs, we have identified specific mechanisms underlying the changes to excitability (firing rate), whole-cell conductance and action-potential waveform.

## Results
**Structural characterisation of the tau samples**. A validated method was used to generate non-fibrillar soluble tau aggregates without the use of aggregation-inducing factors such as heparin[11,22,23]. Incubating all three tau samples at room temperature overnight led to the formation of small soluble aggregates (FL oligomers, C$^{FRAG}$ oligomers, N$^{FRAG}$ soluble aggregates; Fig. 1a), as shown by negative-stain transmission electron microscopy (TEM) for both truncations and the full-length

version of tau. However, there are likely to be structural differences in the aggregate conformers; C$^{FRAG}$-tau is more likely to form granular shaped oligomeric aggregates similar to those of FL-oTau (Fig. 1b and refs. [11,21,22]). Conversely, N$^{FRAG}$-tau is more likely to form more amorphous, irregular-shaped, non-amyloidogenic aggregates, in agreement with ref. [24]. For each variant, monomeric controls (incubated for the same duration but at 4 °C) did not form aggregates (Fig. 1b).

**Removing the N-terminal region of tau (aa 1–123) abolishes the effects of tau on action-potential waveform and input resistance, but the increase in firing rate remains**. We first compared the effects of C$^{FRAG}$-oTau (aa 124–444) to FL-oTau (with the vehicle as control). There was no significant difference in the electrophysiological parameters of the recorded neurons measured at time 0 (whole-cell breakthrough) between all three of the experimental conditions. Therefore, the initial quality of recordings and neuronal properties were comparable. In control neurons, there were no significant changes to any of the measured parameters across the duration of recordings (40 min, Fig. 2 and Table 1). There were also no changes to the resting membrane potential for any of the experimental conditions across the duration of recordings (Fig. 2a, b and Table 1). FL-oTau had comparable effects to those previously reported in ref. [11] increasing input resistance, increasing neuronal firing rate (a correlate of neuronal excitability) and changing the action-potential waveform (reducing height, amplitude relative to baseline and increasing width). These effects were specific to oligomers, with monomeric FL-oTau having no significant effects[11] and are not the result of the tau oligomers blocking the tip of the recording pipette, as a number of controls have previously been carried out to eliminate this possibility[11,19,20].

The increase in firing rate induced by FL-oTau was previously assumed to be the result of the increase in input resistance (Fig. 2c–e) as less current will be required to produce the same voltage change. However, although C$^{FRAG}$-oTau had no significant effect on input resistance (Fig. 2c), it still significantly increased the firing rate (induced by current injection) suggesting that the effects on firing rate can occur independently of changes in input resistance (Fig. 2c–e). In contrast to FL-oTau, which in agreement with ref. [11], significantly increased action-potential width and decreased action-potential height, the introduction of C$^{FRAG}$-oTau had no significant effects on the action-potential waveform (Fig. 2d–g). Thus, loss of the first 123 amino acids from tau removes the effects on input resistance and action-potential waveform. These effects could either be a direct effect of the N-terminal sequence or could result from a change in the structural conformation of the oligomers, which could differ following truncation.

**A reduction in the rheobase current induced by C$^{FRAG}$-oTau is consistent with its effects on firing rate**. To evaluate the effects on neuronal excitability in more detail, we examined the rheobase (the minimal current required to elicit an action potential). The rheobase was measured by injecting a current ramp (see "Methods"). There was no significant difference in the rheobase in control (vehicle) vs FL or C$^{FRAG}$-oTau-injected neurons at time 0 (Table 1 and Fig. 3). In control neurons, there was no significant difference in the rheobase current over the duration of the recording. However, when either FL or C$^{FRAG}$-oTau were introduced, the rheobase was significantly reduced (Table 1 and Fig. 3). This is consistent with the observed increase in firing rate in response to fluctuating noisy current injection (Fig. 2f, g).

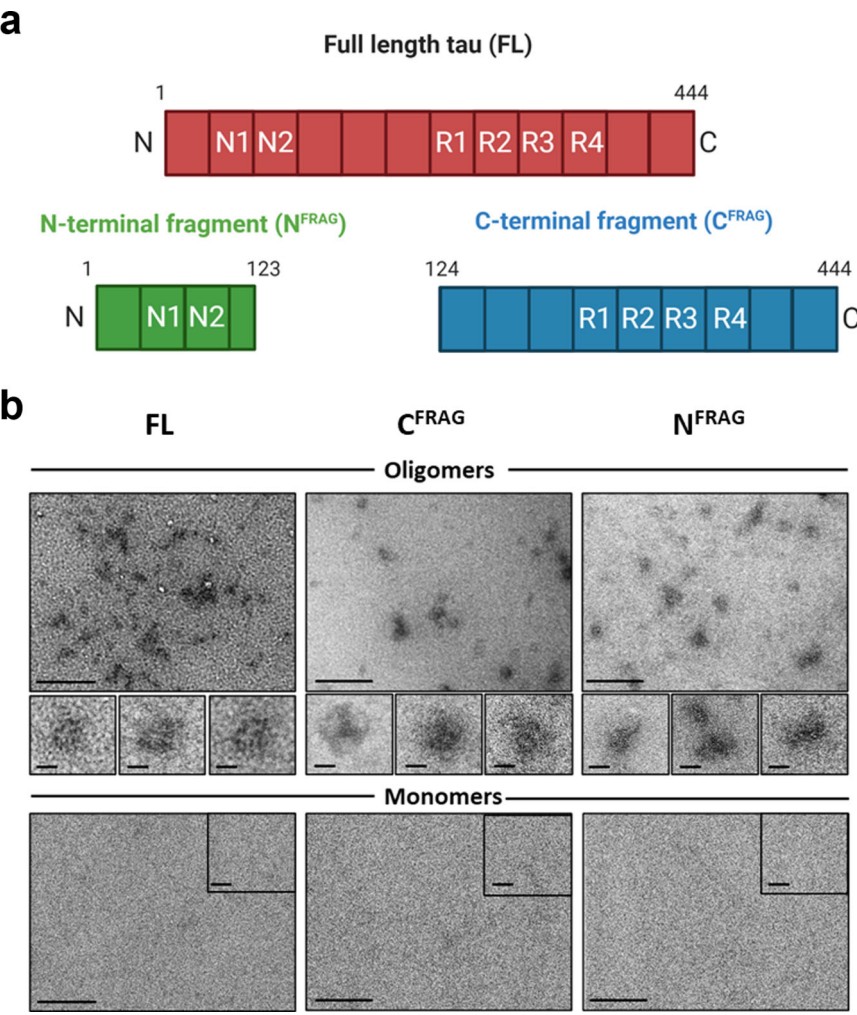

**Fig. 1 Structure and characterisation of the tau truncations. a** Schematic illustration of the recombinant tau constructs used in this study. Full-length (FL) tau (441) has two N-terminal repeat sequences (N1, N2) and four microtubule-binding pseudorepeat domains (R1–R4). Two trunctions were tested. The first is an N-terminal truncated form of FL tau (termed $C^{FRAG}$), which consists of amino acids 124–444. The second is the N-terminal fragment that is removed by the $C^{FRAG}$ truncation (termed $N^{FRAG}$), which consists of amino acids 1–123, including both of the N-terminal repeat sequences. **b** All three versions of tau (FL, $C^{FRAG}$ and $N^{FRAG}$) were visualised as monomers, and after undergoing the oligomerisation protocol. Overnight incubation at room temperature in the absence of any aggregation inducer (e.g., heparin) allowed each protein variant to form small, soluble tau aggregates, as monitored by negative-stain transmission electron microscopy. While FL and $C^{FRAG}$ both formed granular oligomer-like aggregates, the N-terminal fragment polymerised into amorphous structures with no consistent shape. Insets: regions at higher magnification to illustrate aggregate shape. The monomer preparations did not form any visible aggregates. Scale bars = 200 nm and insets 20 nm.

**$C^{FRAG}$-tau oligomers reduce the current needed to elicit APs.** We hypothesised that the decrease in rheobase current might be due to a smaller difference between the resting membrane potential and spike initiation threshold (value of membrane potential that needs to be surpassed to fire an action potential). Since there is no change in resting membrane potential, this could only occur if there was a hyperpolarising shift in the spike initiation threshold. The dynamic IV method[25–27] provides a protocol to accurately parameterise neurons, including spike initiation threshold (see "Methods"). In ref. [11], it was not possible to map changes in neuronal parameters over time using the dynamic IV method, as it is not very effective if there are significant changes in the action-potential waveform. However, as $C^{FRAG}$-oTau has no effect on the action-potential waveform, it is possible to use this method to measure the spike initiation threshold. There was no significant difference between the spike initiation threshold in control neurons (vehicle) vs neurons receiving in $C^{FRAG}$-oTau at 0 min. In control neurons, there was no significant difference in spike initiation threshold after 40 min (recording duration), whereas for neurons where

$C^{FRAG}$-oTau was introduced, spike initiation threshold was significantly hyperpolarised. Although the reduction in spike initiation threshold was only ~4 mV, across a network this would cause a large shift in excitability. We also noted an increase in spike-onset sharpness, which is a measure of how narrow the voltage range within which a spike starts to initiate, as well as how sharply the spike begins to rise (sharper DIV curve). In control neurons, there was no significant difference in spike onset over the duration of the recording, whereas for neurons where $C^{FRAG}$-oTau was introduced, the onset was increased (steeper, reflecting increased excitability). It can be noted that though the spike-onset range was broader at 40 min, this change (unlike for the spike-threshold parameter VT) had no significant effect on the firing rate (see Fig. 4d).

**$N^{FRAG}$-tau aggregates generate a rapid increase in input resistance and change action-potential waveform.** As $C^{FRAG}$-oTau did not replicate the changes to input resistance and action-potential waveform observed with FL-oTau, we next introduced

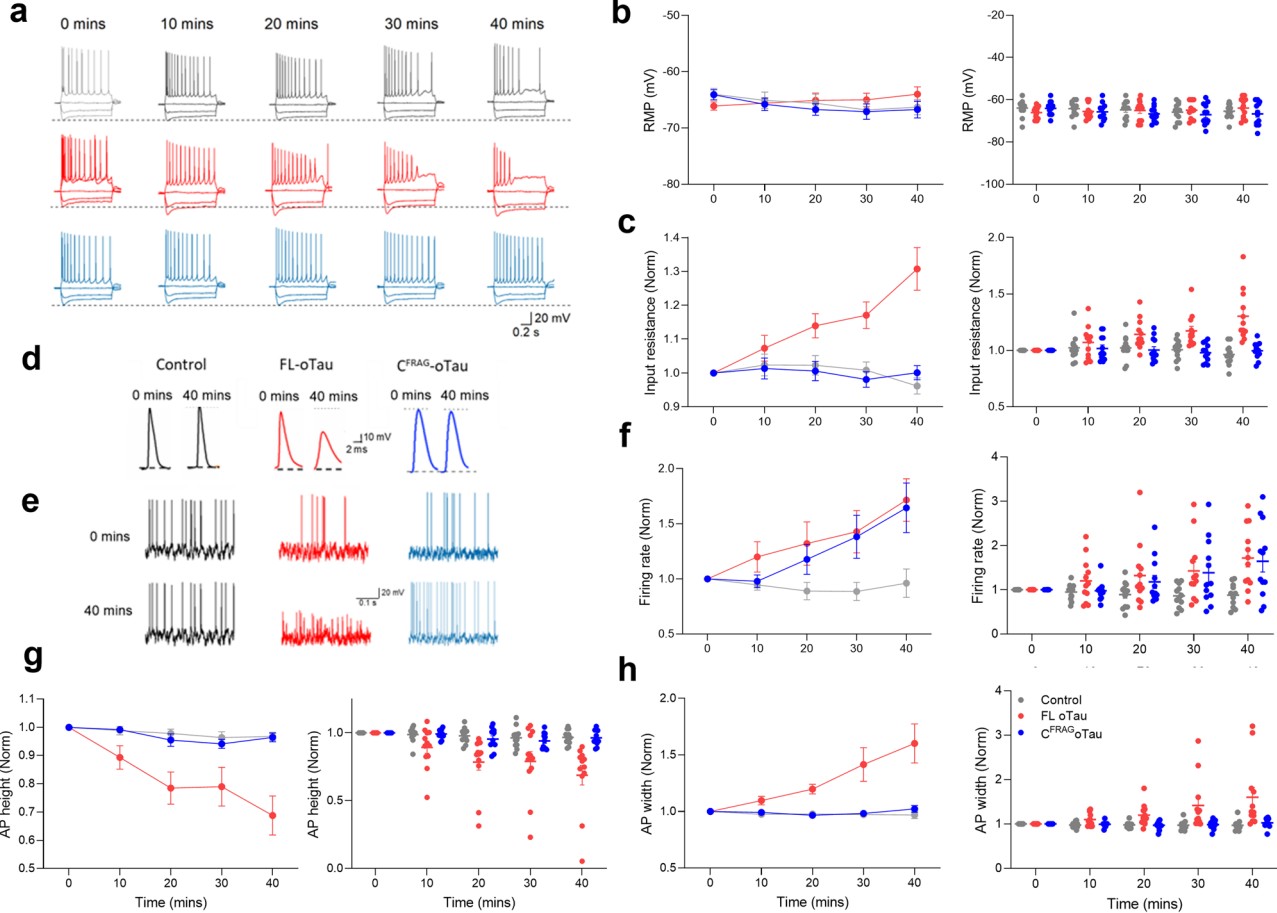

**Fig. 2 The effects of FL-oTau on input resistance and action-potential waveform are abolished by N-terminal truncation, although the effects on firing rate remain. a** Standard current–voltage responses for control neurons (grey; $n = 12$ cells, eight animals) in which vehicle was injected, for neurons in which FL-oTau was introduced (red; $n = 12$ cells, eight animals) and for neurons where $C^{FRAG}$-oTau (blue; $n = 12$ cells, seven animals) was introduced. The responses are shown over the 40-min recording period. There were no significant changes with control neurons to any of the measured parameters (0 vs 40 min, peak input resistance $P = 0.1455$, firing rate $P = 0.1831$, action-potential amplitude $P = 0.0732$ or action-potential width $P = 0.2266$). However, there were clear changes to neuronal properties in the response to FL-oTau and $C^{FRAG}$-oTau. Panels **b**, **c**, **f**, **g** and **h** display both average data (mean ± SEM) as well as individual data points for each condition over time. **b** Graph plotting resting membrane potential against time. There was no change over the period of recording for any of the three experimental conditions (0 vs 40 min; vehicle $P = 0.0898$, FL-oTau and $C^{FRAG}$-oTau, $P = 0.0898$, 0.0859 and 0.0801, respectively). **c** Graph plotting input resistance against time. FL-oTau significantly increased input resistance (at 40 min, input resistance was increased to 130 ± 6.3% of the value at time 0, whole-cell breakthrough, $P = 0.0005$), whereas $C^{FRAG}$-oTau had no effect (at 40 min, the input resistance was 100 ± 2.1% of the value at time 0, $P = 0.9263$). **d** Example action-potential (AP) waveforms for the three experimental conditions recorded at time 0 (whole-cell breakthrough) and after 40 min. Neither vehicle nor $C^{FRAG}$-oTau had any effect on the action-potential waveform. FL-oTau reduced the height (amplitude measured at the peak relative to baseline) and increased the width of the AP. **e** Example membrane-potential responses to naturalistic current injection at time 0 (after whole-cell breakthrough) and after 40 min of recording for each of the three conditions (vehicle, FL-oTau and $C^{FRAG}$-oTau). For FL-oTau, there was a reduction in action-potential height coupled with an increase in the firing rate. $C^{FRAG}$-oTau had no effect on action-potential height but did significantly increase the firing rate. The vehicle had no effect. **f** Graph plotting normalised firing rate (normalised to time 0) against time. Both FL- and $C^{FRAG}$-tau oligomers produce a significant increase in firing rate (FL-oTau: at 40 min, the firing rate measured from the voltage response to naturalistic current injection, was 171 ± 19.3% of that at time 0, $P = 0.02$; $C^{FRAG}$-oTau: at 40 min, the firing rate was 164 ± 22.5% of that at time 0, $P = 0.0132$). No change was observed with the vehicle. **g** Graph plotting action-potential height against time and (**h**) action-potential width against time. A reduction in action-potential amplitude and increase in action-potential width are produced by FL-oTau (action-potential amplitude at 40 min significantly ($P = 0.001$) reduced to 70 ± 7.4% of that at time 0 min and action-potential width at 40 min significantly ($P = 0.0352$) increased to 134 ± 17.2% of that at time 0 min) but not by $C^{FRAG}$-oTau (action-potential amplitude at 40 min 91 ± 1.6% of that at time 0 min, $P = 0.0791$ or action-potential width at 40 min 102 ± 3.0% of that at time 0 min $P = 0.4746$). Error bars represent the standard error of the mean (SEM).

$N^{FRAG}$-tau (aa 1–123) and measured changes in neuronal parameters to determine if this fragment of tau could produce these effects. Unexpectedly, in preliminary experiments, we observed a large increase in input resistance induced by $N^{FRAG}$-tau within the first 5 min of recording (Fig. 5a). This effect occurred without a change in the bridge balance confirming that the tau fragment is not blocking the patch pipette tip (Supplementary Fig. 8). It has

previously been shown that FL-oTau neither aggregates in the end of the pipette tip nor impedes voltage measurements through re-patching experiments[11]. However, this rapid change could be due to $N^{FRAG}$-tau forming aggregates close to the introduction site in the soma and reducing the apparent electrotonic size of the soma. By fitting the voltage step (exponential) we extracted the time constant and used this to calculate cellular capacitance using

**Table 1 Electrophysiological parameters measured for CA1 neurons with either vehicle, C$^{FRAG}$ or N$^{FRAG}$ introduced.**

| Parameter | Vehicle/ control (n = 12) | | | | Full-length tau (n = 12) | | | | C$^{FRAG}$-oTau (n = 12) | | | |
|---|---|---|---|---|---|---|---|---|---|---|---|---|
| | 0 min | | 40 min | | 0 min | | 40 min | | 0 min | | 40 min | |
| | Mean | SEM | Mean | SEM | Mean | SEM | Mean | SD | Mean | SEM | Mean | SEM |
| RMP (mV) | −64 | ±1.07 | −66 | ±1.36 | −66 | ±0.75 | −64 | ±1.30 | −64 | ±0.92 | −67 | ±1.52 |
| Input resistance (MΩ) | 204 | ±18.6 | 197 | ±20.1 | 165 | ±6.29 | 218 | ±16.1 | 178 | ±13.4 | 177 | ±11.7 |
| Firing rate (Hz) | 1.73 | ±0.4 | 1.63 | ±0.45 | 1.25 | ±0.3 | 1.73 | ±0.3 | 1.1 | ±0.19 | 1.5 | ±0.21 |
| AP height (mV) | 85 | ±2.52 | 83 | ±2.86 | 87 | ±3.42 | 63 | ±7.48 | 89 | ±2.19 | 86 | ±2.45 |
| AP width (ms) | 1.8 | ±0.09 | 1.7 | ±0.08 | 1.8 | ±0.05 | 2.4 | ±0.34 | 1.8 | ±0.06 | 1.9 | ±0.06 |
| Rheobase (pA) | 61 | ±3.47 | 60 | ±4.51 | 68 | ±5.39 | 47 | ±7.42 | 79 | ±7.74 | 56 | ±8.18 |

AP action potential, FR firing rate, IR input resistance, RMP resting membrane potential.
Recordings were made from hippocampal CA1 neurons. There was no significant difference between any of the measured parameters at 0 min (whole-cell breakthrough), using Kruskal-Wallis ANOVAs (see text for P values). Comparisons were then made between 0 and 40 min for each of the conditions using non-parametric Wilcoxon signed-rank tests (see text for P values). Data are shown as mean and SEM.

the following equation:

$$\text{Time constant}\,(T) = \text{Input resistance}\,(R_{in}) \times \text{Capacitance}\,(C)$$

$$(1)$$

By 5 min, the time constant was significantly reduced, reflecting a change in whole-cell capacitance. To test the possibility that this rapid change to whole-cell resistance and capacitance could be due to the aggregation of the N$^{FRAG}$-tau in the soma impeding current flow, we performed a subset of experiments where two simultaneous whole-cell patch-clamp recordings were made from the soma of the same CA1 pyramidal neuron (Fig. 5b–f, see "Methods"). We injected current only via pipette 1 and then measured the ratio of the voltage responses in pipette 1 and pipette 2 (voltage pipette 2/voltage pipette 1) (Fig. 5c). We hypothesised that if N$^{FRAG}$-tau was injected via pipette 1, there will be an increase in the whole-cell resistance, less current will leak out from the neuron and thus the voltage response measured from both pipettes will be increased without a change in the ratio. If however the N$^{FRAG}$-tau aggregates in the soma and interferes with the current flow between the pipettes then the ratio of the voltage response between the pipettes will be reduced as less current will reach pipette 2.

A 200 pA (1 s) hyperpolarising step was only injected via pipette (1) and voltage responses were measured from both pipettes (Fig. 5c, pipettes 1 and 2). As the current needs to travel from pipette 1 to pipette 2, the voltage response was always slightly smaller in pipette 2 than in pipette 1. In cells where both pipettes contained vehicle, there was no change in the relative amplitude of the voltage responses between 0 min (whole-cell breakthrough) and 5 min (Fig. 5d). However, when N$^{FRAG}$-oTau was introduced via pipette 1, there was a large increase in the voltage response measured by pipette 1 as a result of the increase in cell resistance (Fig. 5e). However, this was not reflected by an increase in the amplitude of the voltage step measured by pipette 2 (Fig. 5e, f). Thus, the ratio of voltage pipette 2/voltage pipette 1 was significantly reduced (Fig. 5e, f). This result is consistent with N$^{FRAG}$-tau accumulating in the soma, impeding the flow of current between the two pipettes, resulting in an 'increase in the apparent input resistance and a decrease in the observed capacitance due to the cell appearing electronically smaller.

**N$^{FRAG}$-Tau-mediated changes to action-potential waveform are independent of the aggregation state.** We have demonstrated that the FL-oTau mediated increase in input resistance and changes to the action-potential waveform occurred at later time points than we observed with N$^{FRAG}$-tau. This may be because the FL-oTau does not aggregate as rapidly as N$^{FRAG}$-tau and thus takes longer to have its effects. To examine the effects of N$^{FRAG}$-tau in more detail we used a lower concentration (133 nM) which may mimic the slower effects of FL-oTau. There were no effects of 133 nM N$^{FRAG}$-tau on resting membrane potential, input resistance, firing rate or the rheobase current (Fig. 6a–d). However, there were significant changes to the action-potential waveform (increased width and decreased height, Fig. 6e–g). These changes were similar to that observed with FL-oTau in this study and in ref. [11]. This was unexpected as the N-terminal fragment would be predicted to form structurally different aggregates to those formed by FL-tau oligomers. We then repeated the experiments with the monomeric version of N$^{FRAG}$-tau and found no changes in resting membrane potential, input resistance, firing rate or rheobase current (Fig. 6a–d). However, a comparable change to the action-potential waveform was observed in the aggregated form of the fragment (Fig. 6e–g). Given that the oTau-induced changes to action potential

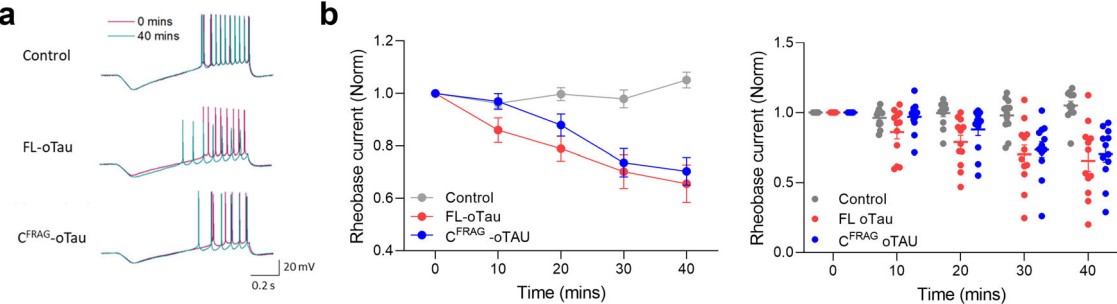

**Fig. 3 FL-oTau and C$^{FRAG}$-oTau reduce the rheobase (minimum current required to fire an action potential).** In the same recordings that are illustrated in Fig. 2, the rheobase current (minimal current to evoke an AP) was determined by injecting a current ramp (−50 to 200 pA) and measuring the minimum current required to fire an action potential. **a** Example voltage responses to ramp current injections in control (vehicle) neurons where FL-oTau was injected and neurons where C$^{FRAG}$-oTau was introduced. The voltage responses are shown at zero min (whole-cell breakthrough) and after 40 min of recording (the traces are superimposed). There was no change in control recordings (vehicle) but there is a clear negative shift (reduction in current required to fire an action potential) for both tau species. **b** Graph plotting the mean rheobase current for control (vehicle), FL and C$^{FRAG}$-tau oligomers against time (mean ± SEM) and the corresponding individual data points. There was no significant difference in rheobase in control, FL or C$^{FRAG}$ neurons at time 0 (whole-cell breakthrough) using a Kruskal–Wallis one-way ANOVA (at 0 min, the mean rheobase in control was 61 ± 3.47 pA, with FL the mean rheobase was 68 ± 5.39 pA and with C$^{FRAG}$ the mean rheobase was 79 ± 7.74 pA, $P = 0.1289$). In control neurons, there was no significant difference in the rheobase current over the duration of the recording (at 40 min, the rheobase was 105 ± 3% of the value at time 0 min, $P = 0.0830$, Table 1). However, both FL and C$^{FRAG}$-tau oligomers significantly reduced the current needed to elicit an action potential. For FL-oTau, at 40 min the rheobase was 66 ± 7% of the value at time 0 min ($P = 0.0015$). When C$^{FRAG}$-tau was introduced, at 40 min the rheobase was 70 ± 5% of the value at time 0 min ($P = 0.0005$, Table 1). Error bars represent the standard error of the mean (SEM).

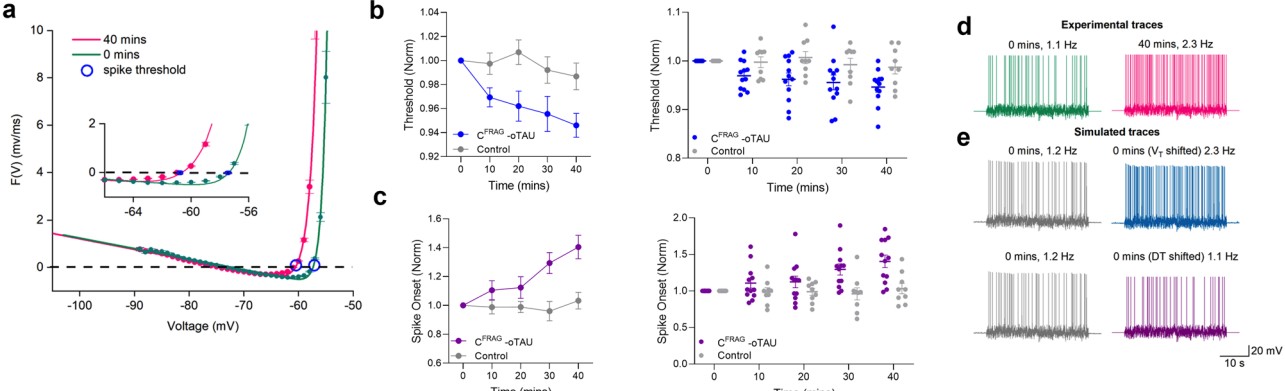

**Fig. 4 oTau-mediated hyperpolarising shift in spike initiation threshold.** For the dynamic I–V protocol, a naturalistic current is injected into the cell, and the recorded voltage is then used to extract a set of parameters (resting potential $E$, time constant $\tau$ and spike-threshold voltage $VT$; see "Methods" for more details). The second point where the DIV curve crosses 0 mV/ms is the spike initiation threshold. **a** Representative examples of dynamic IV curves, for a neuron with the N-terminal truncation of tau (C$^{FRAG}$-oTau) introduced, generated at 0 min (green) and after 40 min (pink). A hyperpolarising shift in spike initiation threshold of 4 mV (more negative) can be observed (circles on the graph). Panels **b** and **c** display both average data (mean ± SEM) as well as individual data points for each condition over time. **b** The mean normalised spike-threshold data (to value at time 0) is plotted against time. There was no significant difference between the spike initiation threshold in control vs C$^{FRAG}$ neurons at time 0 (whole-cell breakthrough), thus the recordings were comparable (mean threshold in control neurons was −53.82 ± 1.46 mV, $n = 12$ cells, compared to the mean threshold in C$^{FRAG}$ neurons which was −53.7 ± 1.64 mV, $n = 12$ cells, $P = 0.5185$). In control neurons, there was no significant difference in spike threshold over the duration of the recording (at 40 min, the spike threshold was 101.3 ± 1.1% of the value at time 0 min, $P = 0.4766$). Whereas for neurons where C$^{FRAG}$ was introduced, the spike initiation threshold was significantly reduced (more negative) at 40 min was compared to 0 min (at 40 min, the spike threshold was 94.5 ± 0.9% of the value at time 0 min, $P = 0.0010$). **c** Graph plotting spike onset against time. There was no significant difference between the onset in control vs C$^{FRAG}$ neurons at time 0 (whole-cell breakthrough), thus the recordings were comparable in quality and stability (mean spike onset in control neurons was 1.01 ± 0.09 mV, $n = 12$ cells, compared to the mean threshold in C$^{FRAG}$ neurons which was 0.98 ± 0.13 mV, $n = 12$ cells, $P = 0.1285$). In control neurons, there was no significant difference in spike onset over the duration of the recording (at 40 min, the spike onset was 103.2 ± 5.7% of the value at time 0 min, $P = 0.9102$). Whereas for neurons where 444 nM aggregated C$^{FRAG}$-tau was introduced, the onset was increased (steeper, reflecting increased excitability) at 40 min was compared to 0 min (at 40 min, the spike onset was 140.4 ± 8.2% of the value at time 0 min, $P = 0.0015$). The increase in spike onset observed in neurons where C$^{FRAG}$-oTau was introduced reflects faster spike generation after the threshold was reached. **d** Membrane-potential responses to naturalistic current injection for the neuron analysed in (**a**) demonstrates an increase in firing rate mediated by the introduction of C$^{FRAG}$-tau oligomers. **e** Simulated (refractory integrate and fire model) membrane-potential responses produced from the extracted neuronal parameters at time 0 min (green). The firing rate in the simulation is a good match to what was measured experimentally. The spike threshold was shifted by −4 mV and the simulation re-run (blue). The increase in firing rate was comparable to the experimental data at 40 min (**d**). The same procedure was performed, but only with a shift in the spike onset. This had little effect on the firing rate, thus confirming that the increase in firing rate is predominantly mediated by a shift in spike threshold. Error bars represent the standard error of the mean (SEM).

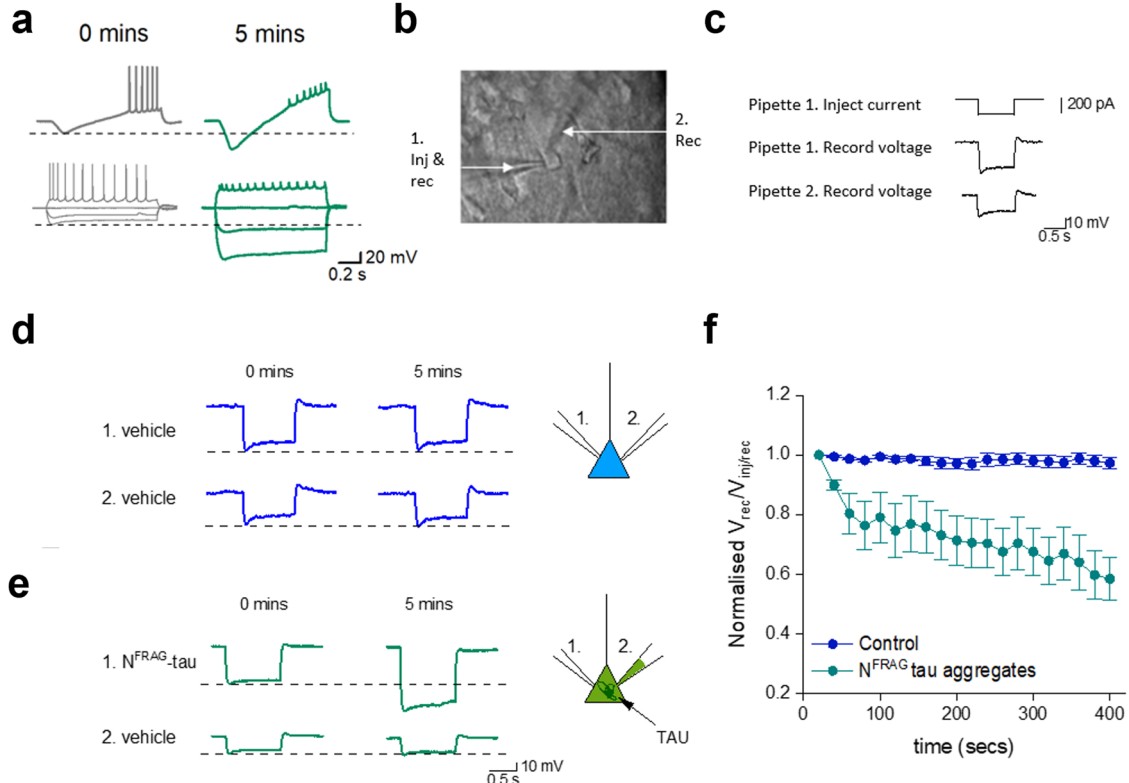

**Fig. 5 The rapid increase in input resistance mediated by N^FRAG-tau (444 nM) is due to aggregation in the soma impeding current flow. a** Representative examples of voltage responses to ramp current injection and voltage responses to step current injections at 0 min (whole-cell breakthrough) and after 5 min for neurons with N^FRAG-tau (444 nM) injected. N^FRAG-tau rapidly induces a large increase in input resistance and a reduction in spike height. At 5 min, the input resistance was 169 ± 24.2% of that at 0 min (whole-cell breakthrough), the time constant was reduced to 77 ± 3.2% of that at 0 min and this change in time constant reflected a change in whole-cell capacitance (by 5 min the capacitance was reduced to 53 ± 5.4% of that at 0 min ($n = 4$ cells, four animals). **b** Bright-field image of a hippocampal CA1 pyramidal neuron showing two patch pipettes simultaneously recording from the soma. In this example, pipette 1 was used to inject current and record the voltage response, whereas pipette 2 was used only for recording the voltage responses. **c** Schematic of the stimulation protocol. A 200 pA hyperpolarising current step was injected via pipette 1, and the voltage response was recorded from both pipettes (1 and 2). **d** An example experiment where both pipettes contained vehicle. There was no change in the relative amplitude of the voltage responses between 0 min (whole-cell breakthrough) and 5 min for pipette 1 and pipette 2 (the relative response was 97 ± 1.7% of the response at 0 min, $P = 0.1484$, $n = 8$ cells, four animals). **e** An example experiment where 444 nM N^FRAG-tau was introduced via pipette 1. There is a clear increase in the voltage response measured in pipette 1 as a result of an increase in input resistance. However, there is no increase in the voltage response in pipette 2. Thus, the relative ratio of voltage responses between the two pipettes has fallen, indicative of impeded current flow in the soma (at 5 min the relative response was 58.4 ± 7% of the response at 0 min, $P = 0.0078$, $n = 8$ cells, four animals). **f** Graph plotting the mean normalised ratio of voltage responses (pipette 2/pipette 1) for recordings where both pipettes contained vehicle (control, $n = 8$ cells) and when pipette 1 contained 444 nM N^FRAG-tau ($n = 8$ cells). There is a significant reduction in the ratio of voltage responses when 444 nM N^FRAG-tau is injected, suggesting that less current reaches pipette 2. Error bars represent standard error of the mean (SEM).

waveform that occur with full-length tau[11] also occurred with the N-terminal fragment (N^FRAG-Tau), both as a monomer and as an aggregate, it is likely that there is a sequence within the 1–123 amino acid region that directly interacts with a cellular component, rather than being an effect of the quaternary structure of the aggregates.

**FL-oTau alters somatic voltage-gated sodium channel half activation and slows the rate of activation.** Having demonstrated changes to the action-potential waveform and a hyperpolarising shift in spike threshold, making the neurons more excitable, we next investigated whether these changes were mediated by interactions of tau with voltage-gated sodium channels. FL-oTau (444 nM) was used in these experiments to evaluate whether both or either of its observed effects (on action-potential waveform and firing rate) could be a result of an interaction with sodium (NaV) channels. FL-oTau was compared against control (vehicle). Recording sodium channel currents in

hippocampal neurons in acute brain slices is challenging due to problems with space clamp, given the size of the neurons and the large amplitude and speed of the currents. One way to effectively clamp the currents is to electronically isolate the soma. We followed the protocol outlined in ref. [28], using a short depolarising pre-pulse to inactivate the axonal sodium channels, leaving only the somatic channels to be activated (see "Methods"). It was then possible to achieve a reasonable voltage clamp (Fig. 7a). Normalised conductances were plotted against voltage and fitted with a Boltzmann function. The fit (Fig. 7b, c) was used to extract the half-activation voltage and the rate constant of activation (indicative of the rate of rise in conductance relative to the change in voltage). The half activation was stable in control recordings, whereas in neurons that had FL-oTau 444 nM introduced, by 20 min the half activation had shifted significantly in a negative direction to activate ~4 mV earlier, reflective of an increase in excitability (Fig. 7). This could account for the change in the spike threshold that was observed with C^FRAG-oTau. We also extracted the rate constant of activation; a smaller rate constant would

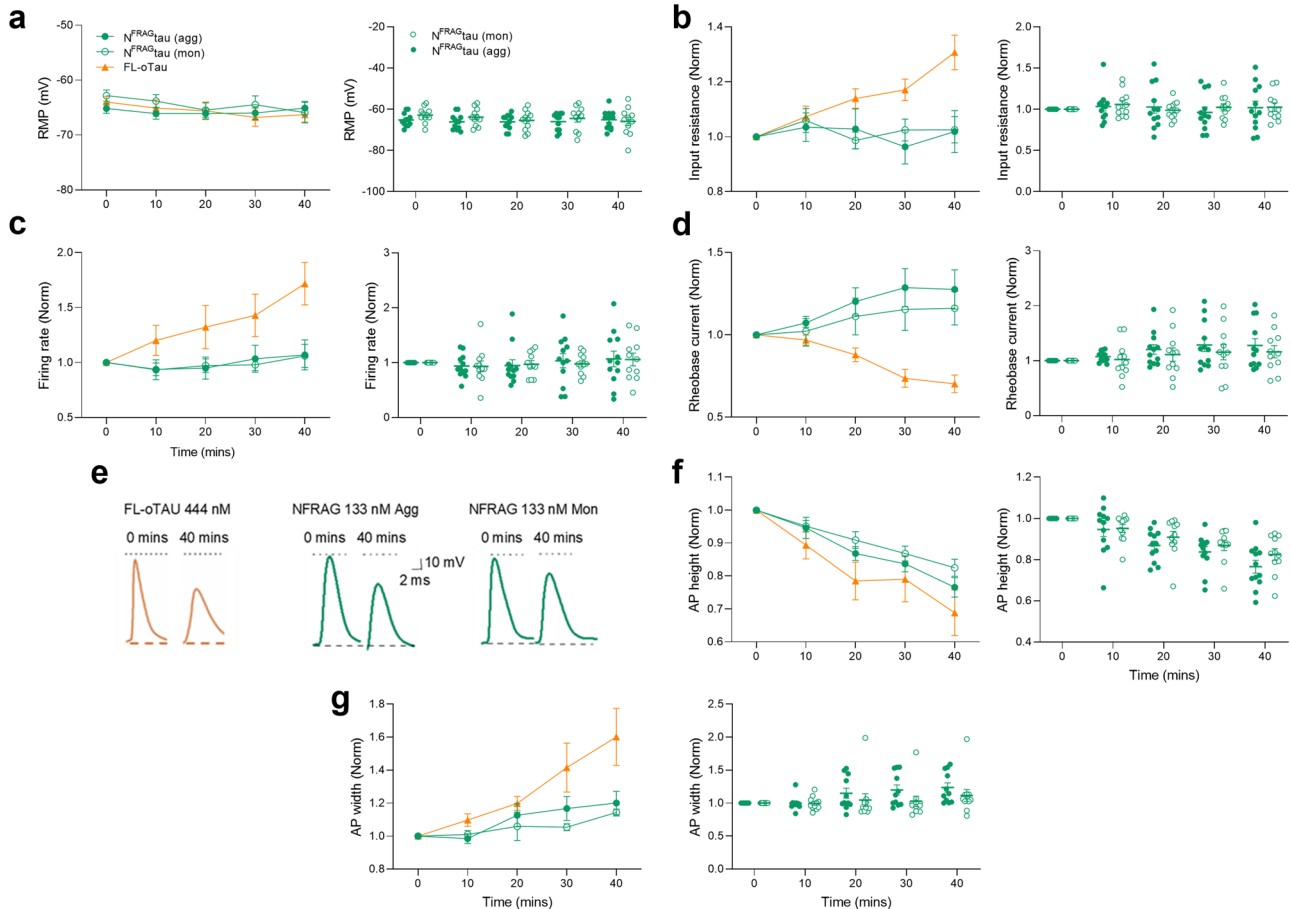

**Fig. 6 The effects of FL-oTau on action-potential waveform are reproduced by lower concentrations of N$^{FRAG}$-tau aggregates and monomers. a–d, f and g** display both average data (mean ± SEM) as well as individual data points for each condition over time. **a** Graph showing that there is no change to the resting membrane potential over the period of recording for any of the conditions (133 nM N$^{FRAG}$-tau monomers or aggregates; N$^{FRAG}$-tau aggregates: at 40 min, the resting membrane potential was 99 ± 1.0% of that at time 0, $P = 0.9609$, $n = 12$ cells, seven animals; N$^{FRAG}$-tau monomers at 40 min, the resting membrane potential was 104 ± 2.4% of that at time 0, $P = 0.0938$, $n = 12$ cells, six animals. **b** Graph showing that there is no change to the input resistance over the period of recording for 133 nM N$^{FRAG}$-tau aggregates or monomers (N$^{FRAG}$-tau aggregates: at 40 min, the peak input resistance, measured from the voltage response to step current injection, was 102 ± 7.7% of that at time 0, $P = 0.9531$; N$^{FRAG}$-tau monomers: at 40 min, the peak input resistance, measured from the voltage response to step current injection, was 102 ± 4.1% of that at time 0, $P = 0.9395$. **c** Graph showing that there is no change to the firing rate over the period of recording for 133 nM N$^{FRAG}$ aggregates or monomers (N$^{FRAG}$-tau aggregates: at 40 min, the firing rate, measured from the voltage response to naturalistic current injection, was 107 ± 13.6% of that at time 0, $P = 0.7471$; N$^{FRAG}$-tau monomers: at 40 min, the firing rate, measured from the voltage response to naturalistic current injection, was 106 ± 10.5% of that at time 0, $P = 0.5068$). **d** Graph showing that there is no change to the rheobase current (minimal current to evoke an AP) over the period of recording for 133 nM N$^{FRAG}$-tau aggregates or monomers (N$^{FRAG}$-tau aggregates: at 40 min, the rheobase was 127 ± 11.9% of that at time 0, $P = 0.1294$; N$^{FRAG}$-tau monomers: at 40 min, the rheobase was 116 ± 10.1% of that at time 0, $P = 0.2402$). **e** Representative examples of average action-potential waveforms from neurons with either 444 nM FL-oTau, 133 nM N$^{FRAG}$-tau aggregates or 133 nM N$^{FRAG}$-tau monomers introduced. In all three conditions, there is a comparable decrease in action-potential height and increase in action-potential width. **f** Graph plotting mean normalised action-potential height (at time 0) against time. There is a comparable decrease in action-potential height for 133 nM N$^{FRAG}$-tau aggregates, monomers and FL-oTau (N$^{FRAG}$-tau aggregates: action-potential amplitude at 40 min was significantly reduced to 76 ± 2.9% of that at time 0 min, $P = 0.0005$; N$^{FRAG}$-tau monomers: action-potential amplitude at 40 min was significantly reduced to 82 ± 2.6% of that at time 0 min, $P = 0.0010$). **g** Graph plotting mean normalised action-potential width (at time 0) against time. There is a comparable increase in action-potential width over the period of recording for 133 nM N$^{FRAG}$-tau aggregates, monomers and FL-oTau (N$^{FRAG}$-tau aggregates: action-potential width at 40 min was significantly increased to 118 ± 7.4% of that at time 0 min, $P = 0.0186$; N$^{FRAG}$-tau 32 monomers: action-potential width at 40 min was significantly increased to 123 ± 2.0 % of that at time 0 min, $P = 0.0303$). Orange lines demonstrate the effects of FL-oTau for comparison (data presented in Fig. 2). AP action potential, FR firing rate, Rin input resistance, resting membrane potential, resting membrane potential). Error bars represent the standard error of the mean (SEM).

indicate a steeper relationship between conductance and voltage. The rate constant of activation was stable in control, whereas in neurons that had FL-oTau introduced, by 20 min it had increased significantly (shallower slope, reflective of a flatter relationship to voltage). This change could explain the slowing of the action-potential rising phase observed in ref. [11] and in this study.

We then implemented a simplified model of the neuronal action potential with general applicability[29] to evaluate whether the conductance changes mediated by FL-oTau could feasibly underly the changes we observed in the action-potential waveform. Our experimental voltage-gated sodium current recordings showed that FL-oTau mediates a reduction of $\bar{g}_{Na}$ to by

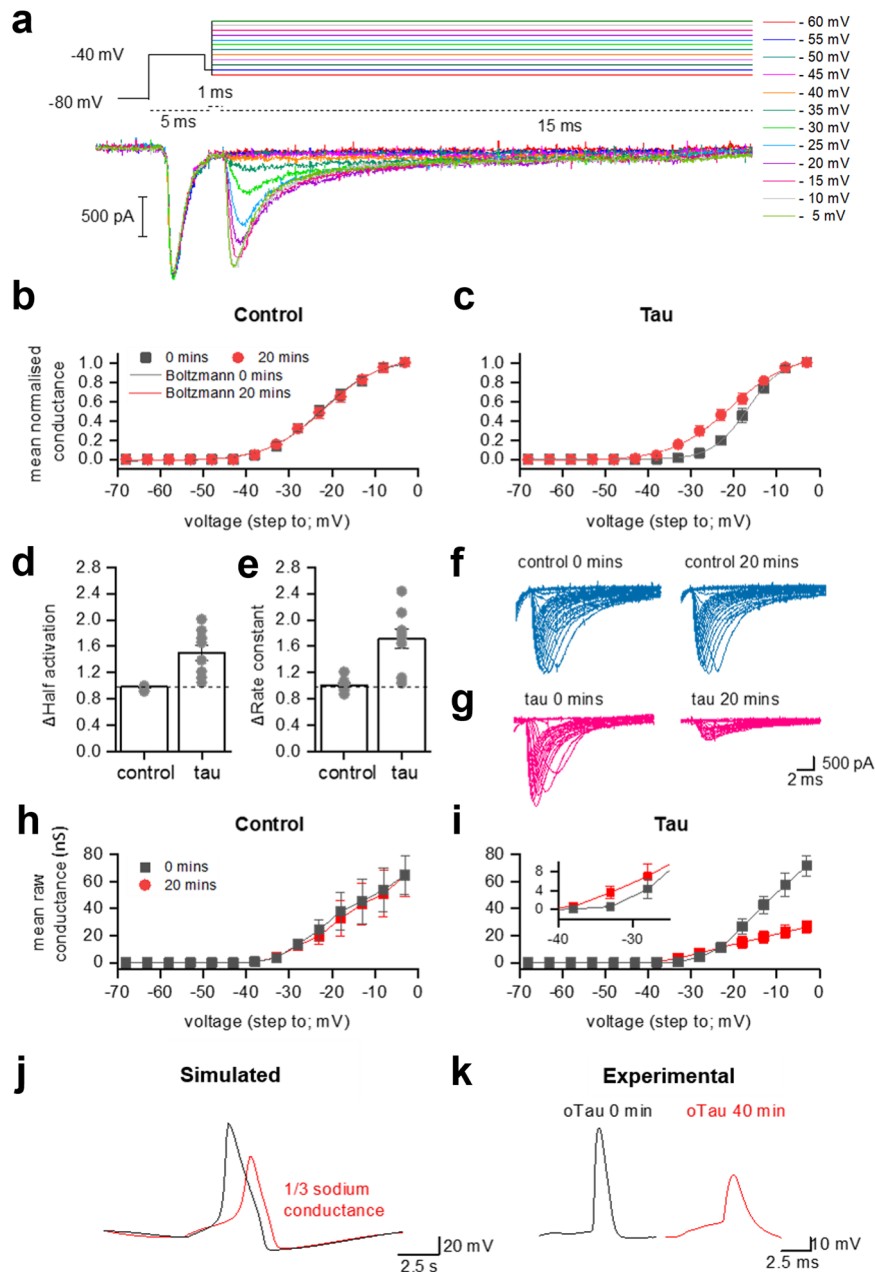

two-thirds of the value at 0 min after 20 min (Fig. 7f–i). Therefore, we first simulated an action potential with $\bar{g}_{Na} = 120\,mS/cm^2$ ([28], Fig. 7j, red) and then again with $\bar{g}_{Na} = 40\,mS/cm^2$ (Fig. 7j, red trace). We found that this reduction in maximal conductance matches the experimental phenotype well (in particular, the changes in action-potential amplitude and speed of rise to threshold; Fig. 7k).

## Discussion

In Alzheimer's disease (AD) and in other tauopathies, tau aggregates into small soluble toxic oligomers, a process that precedes the formation of large insoluble aggregates (NFTs). In AD, NFTs and amyloid-β plaques are the characteristic hallmarks of the disease with post-mortem confirmation of both NFTs and plaques required for definitive diagnosis[6,7,30]. Many studies have shown that the small soluble aggregates (oligomers) formed early in the aggregation pathway are much more bioactive than NFTs pathology[8–10]. These findings have led to the targeting of

oligomers to reduce tau cytotoxicity in disease models and in AD patients. However, the mechanisms of how these oligomers produce neuronal dysfunction are still not fully defined. Here, we have used specific truncations to dissect apart the actions of full-length (FL)-oTau. We have used oligomers formed from N-terminal truncated tau (aa 124–444; $C^{FRAG}$) and soluble aggregates of the non-oligomer forming N-terminal fragment (aa 1–123; $N^{FRAG}$).

It is conceivable that some of the changes in electrophysiological properties that we have observed could result from tau oligomers blocking the tip of the patch pipette and increasing series resistance. In particular the changes in action-potential amplitude and waveform kinetics. In the previous studies[11,19,20] and in this study, we have carried out a number of control experiments that have allowed us to reject this possibility. These controls include monitoring of series resistance throughout recordings and adjusting the bridge balance as required, introducing large concentrations of a control protein (BSA, 20 μM vs 44–144 nM) into neurons and observing no change in series

**Fig. 7 FL-tau oligomers directly modify somatic sodium channel currents recorded from hippocampal CA1 neurons in acute slices. a** Example of the recording protocol (adapted from ref. [27]). A pre-pulse step of 5 ms from (−80 to −40 mV) elicits a current response from axonal (but not somatic) sodium channels. There is a brief step to −55 mV (1 ms) before the steps that are used to evoke the controlled sodium channel currents (15 ms in duration). The steps were given consecutively from −60 V to + 60 mV (5 mV increment) with a 2 s gap between each sweep. Leak currents were subtracted with the P/N protocol (1/4) and the junction potential (+8 mV) was corrected for. **b** In order to look at the full range of the changes that had been observed with the different truncations, FL-oTau (444 nM) was used in these experiments and compared against control. Conductance was calculated using the following equation: conductance $(g)$ = current $(I)/(Vstep − Vreversal)$. The conductance in response to steps up to 0 mV is plotted as a mean (and SEM) for all control cells (vehicle, $n = 7$ cells, six animals). **c** The conductance in response to steps up to 0 mV is plotted as a mean (and SEM) for all cell where FL-oTau (444 nM) was introduced ($n = 7$ cells, five animals) is plotted. For **b** and **c**, the grey line is 0 min and the red 20 min. A Boltzmann fit allows half activation voltage and the time constant (measure of the speed of activation). **d** Mean change in half activation from 0 to 20 min for control and FL-oTau-injected neurons. Data are given relative to the value at time 0. Mean and SEM are shown, and individual data points are overlaid on top. While control (vehicle) cells are stable over time (at 20 min, the half activation was 98 ± 1.0% of that at time 0, $P = 0.2031$), Fl-oTau introduced cells show a significant decrease in half activation (activate at lower voltages; increase in excitability; at 20 min, the half activation was 150 ± 11.7% of that at time 0, $P = 0.0078$), but they also show a doubling of the activation constant (**e**; in control cells, at 20 min, it was 101 ± 3.6% of that at time 0, $P = 0.9375$, whereas in FL-oTau introduced neurons had increased significantly to 171 ± 15.4% of that at time 0, $P = 0.0078$), fitting with the slower rise time of APs observed in ref. [11]. **f**, **g** Example sodium currents from neurons with vehicle introduced (control) and FL-oTau introduced at 0 min and after 20 min demonstrating the reduction in peak current induced by FL-oTau, with no change introduced in control. **h**, **i** Raw conductance data (averaged) to highlight the reduction in maximal conductance for FL-oTau after 20 min. Inset demonstrates the crossover at low voltages, where FL-oTau at 20 min produces a higher conductance, despite the maximal conductance being reduced at higher voltages. **j** A simple Hodgkin–Huxley model of action-potential dynamics was used to predict the effect of reducing the maximal sodium channel conductance on the action-potential waveform. Reducing the maximal sodium channel conductance by two-thirds, as measured experimentally in (**i**) produced similar changes in the action-potential waveform (**j**) to that observed experimentally with FL-oTau (**k**). Error bars represent the standard error of the mean (SEM).

resistance, re-patching neurons (with vehicle in the patch solution) which had been injected with tau oligomers and showing that the changes in action-potential waveform persisted[11]. We have subsequently checked that the open tip resistance of patch pipettes does not change over time (as a result of clogging with tau oligomers). In this study, we have found that it is the smallest fragment of the tau oligomer that has the greatest effects on action waveform, which is the opposite to what would be expected if it was just the tau oligomers non-specifically blocking the pipette tip. We have also previously investigated the effects of alpha-synuclein oligomers on neuronal properties by introducing them via the patch pipette[19]. These oligomers are much larger than the tau oligomers but had no effect on the action-potential amplitude or waveform kinetics[19].

The N-terminal truncation ($C^{FRAG}$) at position 123, has been identified as a major N-terminal cleavage site in tau found in human CSF[21] and a truncated tau construct at a nearby position has been used to study effects on synaptic transmission[31]. We found that $C^{FRAG}$-oTau did not replicate the effects on action-potential kinetics or input resistance that were observed with FL-oTau but did still increase firing rate. This firing rate increase is mediated by a reduction in the minimum current required to elicit an action potential. Using the dynamic IV protocol, we found that $C^{FRAG}$-oTau reduces spike threshold by ~4 mV. Since there was no change in the resting membrane potential, the difference between spike threshold and resting potential was reduced, increasing the likelihood of firing. A hyperpolarisation of spike threshold (~3 mV) has also been previously reported in Tau35 transgenic mice[32]. To confirm our observation further, the parameters extracted from the dynamic IV curve at 0 min were used to simulate voltage responses using the exponential integrate and fire model[25–27,33]. The control firing rate was comparable to experimental values. A reduction in the spike threshold by the experimentally observed reduction (4 mV) was sufficient to increase the firing rate to comparable experimental values with $C^{FRAG}$-oTau. Thus, it is feasible for a change in spike threshold (without any change in input resistance) to mediate the observed changes in firing rate (a correlate of excitability) by $C^{FRAG}$-oTau.

We then investigated the tau fragment that was removed to generate $C^{FRAG}$ (aa 1–123; $N^{FRAG}$). This fragment ($N^{FRAG}$)

makes up a large proportion of N-terminal tau species found in human CSF, and it could therefore represent a marker of tau metabolism[21]. $N^{FRAG}$ does form aggregates but these are likely to differ from the FL-oTau oligomers. When $N^{FRAG}$ was introduced into neurons at equivalent concentrations to FL-oTau (444 nM), it produced a rapid increase in input resistance, reduced action-potential amplitude and slowed action-potential kinetics. These effects occurred within ~5 min. Through measurements of the cell time constant, we found that the apparent capacitance was significantly decreased. This led us to hypothesise that $N^{FRAG}$ was inferring with current flow, making the neuron electrotonically smaller. By making simultaneous dual patch-clamp recordings from the soma of a single neuron, we found that although input resistance was significantly increased, the current flow between the electrodes had significantly decreased. This is consistent with $N^{FRAG}$-tau aggregating in the soma and impeding current flow across the cell.

Introducing $N^{FRAG}$ at lower concentrations (133 nM) changed action-potential height and width without increasing input resistance or changing cell capacitance. Thus, the changes in action-potential waveform were not simply due to increased filtering. Surprisingly, $N^{FRAG}$ monomers produced comparable changes to APs. Thus, the effects of $N^{FRAG}$ are unrelated to its quaternary structure and instead maybe associated with a specific sequence within $N^{FRAG}$ and FL-oTau. Given the nature of the changes in action-potential waveform, we hypothesised that tau might be interacting with voltage-gated sodium channels and altering their function.

Changes in action-potential kinetics and amplitude can readily be measured using whole-cell current clamp[29]. However, recording the underlying sodium currents from neurons in slices using voltage clamp is challenging due to poor space clamp and the large amplitude and rapid rise of currents[34–36]. A large proportion of sodium channels are clustered in the axon initial segment (AIS[35–37]), and their activation voltage can differ from that at the soma[37–39]. Thus, these electrotonically remote sodium channels will generate unclamped currents, which prevent the measurement of controlled responses. To counteract this problem, many studies measurements of sodium channel currents are performed using dissociated or cultured neurons or in cell lines. However, these are less relevant physiologically relevant

because the distribution, density and expression pattern of channels will be different from what is found in situ. An alternative approach is to use nucleated patches, which would provide a better voltage clamp, but the small volume of the patch may alter the amount of tau reaching the channels. It would therefore be difficult to accurately titrate the concentration of tau to make it compatible with the experiments done in whole cells. We have therefore adapted a protocol[28] to make controlled voltage clamp recordings in acute slices. A pre-pulse elicits an uncontrolled voltage response that activates axonal but not somatic sodium channels. As the axonal channels are inactivated, only the sodium channels in the soma contribute to the actual recorded current, allowing for finely controlled voltage clamp of the sodium currents. This protocol also required a number of other steps to produce clamp of isolated sodium channel currents including changing the $Na^+$ concentration inside and outside of the neuron, performing recordings at room temperature and replacing the calcium ions with cobalt to block voltage-gated calcium channels[40]. Those recordings in which $Na^+$ channels were not adequately clamped were readily identifiable and excluded from analysis (Supplementary Fig. 7).

There are four subtypes of voltage-gated sodium channel present in the mammalian central nervous system, NaV1.1, 1.2, 1.3 and 1.6[41]. At the AIS of CA1 pyramidal neurons, NaV1.1, NaV1.2 and NaV1.6 channels have been identified[42–44]. NaV1.6 is highly abundant and has a more hyperpolarized voltage of activation compared with the other sodium channel isoforms, contributing to the lower activation threshold[38]. The sodium channels in the AIS probably contribute little to the clamped currents and this is a possible limitation of the technique. However, NaV1.6 is also expressed in the soma, although at less abundance than at the AIS[45].

FL-oTau had three effects on sodium channel currents: firstly, the half activation was reduced, reflecting the increase in neuronal excitability, consistent with the effects of $C^{FRAG}$-oTau. Secondly, the rate of activation was reduced underlying the slower action-potential rise, consistent with the effects of $N^{FRAG}$-tau. Finally, FL-oTau reduced $\bar{g}_{Na}$ (maximum sodium conductance) by ~two-thirds. To investigate the effects this would have on the AP, we implemented a simplified action-potential model[29]. Reducing $\bar{g}_{Na}$ had effects that matched the experimental data (reduced action-potential amplitude and change to threshold). Thus, this reduction in conductance could account for the changes that we observe experimentally.

From unnormalized conductance plots, we noticed that the rate of conductance increase appeared faster for FL-oTau despite $\bar{g}_{Na}$ being reduced (Fig. 7i). We, therefore, modelled the initial rise of the sodium current using the same approach[33] used to derive the original model:

$$I_{Na} = \left(E_{Na} - E_{rest}\right) * g1 * e^{\left(\frac{V - V1}{\Delta T}\right)} \qquad (2)$$

Where $g1$ is 1 nS and $V1$ is the voltage where the conductance is 1 nS. For FL-oTau-introduced cells, $\Delta T$ was calculated to be stable ~2.65 mV for 0 min and 20 min but $V1$ varied over time. At 0 min, $V1 = -32$ mV and by 20 min, $V1 = -36$ mV, demonstrating that a sodium conductance of 1 nS is achieved at a much lower voltage, thus the cell is more excitable (despite the peak sodium conductance being lower overall, Fig. 7i). In the EIF model, the sodium current is modelled as:

$$\Delta T * gL * e^{\left(\frac{V - VT}{\Delta T}\right)} \qquad (3)$$

and comparing this with the equation for $I_{Na}$ we get:

$$VT = V1 + \Delta T * \log\left(gL * \frac{\Delta T}{g1} * \left(E_{Na} - E_{rest}\right)\right) \qquad (4)$$

Empirically, the first term $V_1$ on the right-hand-side is dependent on parameters that change between 0 and 40 min, whereas the second term does not change significantly between these time steps. Hence, the modelling predicts that the change in the action-potential threshold parameter $V_T$ is largely due to changes in $V_1$ which is the voltage at which the sodium current is 1 nS. The $I_{Na}$ experiments predict that the difference in $VT$ at 0 and 40 min is 4 mV, which fits exactly with the excitability phenotype modelled using $C^{FRAG}$. Despite the differences in temperature for these recordings, these two independent methodologies further support that the tau-mediated shift in excitability comes from a lowering of the action-potential threshold (rather than input resistance changes) and that this change is mediated via a lowering of the sodium current activation. In future studies, it would be interesting to investigate the effects of tau oligomers on sodium channel inactivation. If the inactivation curve is negatively shifted by tau oligomers, this would increase the fraction of sodium channels inactivated at the resting membrane potential and this could also contribute to the reduction in maximal conductance.

It is surprising to note that the difference in sodium conductance at 0 min and 20 min with FL-oTau is dependent on voltage, at low voltages the conductance is bigger at 20 min in FL-oTau but at high voltages it is lower at 20 min in oTau, thus, the effect is non-monotonic. Globally it appears that there is a lower maximal Na conductance at 20 min (and therefore action-potential height is reduced), but there is a small region at lower voltages where the conductance is higher for FL-oTau which means the model is more excitable. This means the profile of the $I_{Na}$ activation has been changed by oTau; it is not just an issue of different maximal conductance magnitudes.

The simplest explanation for these changes to sodium conductance is that a specific $N^{FRAG}$-tau sequence binds to the sodium channel. The mammalian sodium channel is a molecular complex of an ~2000 amino acid α-subunit, which contains the pore and drug interaction sites, and smaller β-subunits, which modulate membrane expression[46,47]. The α-subunit is divided into four domains each with six segments (S1–S6). S1–S4 form the voltage sensing domain[48] and S5–S6 form the pore-forming domain. Upon depolarisation, S6 segments move leading to channel opening via the P-loop between S5 and S6 which forms the channel pore[41,49]. The S4 segments contain a high proportion of positively charged amino acid residues, making it responsive to changes in membrane potential[50]. It is interesting to speculate on where the tau oligomers could bind to sodium channels, changing the coupling between depolarisation and pore opening. Many studies have shown that changes in the S4 segments, associated linkers and cytoplasmic loops can all change the activation of the channels[41,49].

This work has shown that by modifying the tau molecule it is possible to dissect apart the multiple effects it has on neuronal function (Table 2) and identify the molecular targets that tau might bind to. This not only provides information about the mechanisms of action of tau but also raises interesting possibilities. Since the truncations used in this study have been found in the CSF of AD patients, then the effects of tau will depend on the mixture of these molecules inside neurons. Thus, if tau is primarily present as the $C^{FRAG}$ then it will affect excitability and not change action-potential waveform. However, if the $N^{FRAG}$ accumulates in the soma it will have large effects on neuronal integration and action-potential waveform. It would also be interesting to identify whether these effects are specific to specific

| Parameters | FL- oTau (444 nM) | C$^{FRAG}$ -oTau (444 nM) | N$^{FRAG}$-oTau (444 nM) | N$^{FRAG}$-oTau (133 nM) | N$^{FRAG}$-Tau (133 nM) | Potential mechanism |
|---|---|---|---|---|---|---|
| AP height | ↓ | – | ↓ | ↓ | ↓ | Slower activation of voltage-gated sodium channels |
| AP width | ↑ | – | ↑ | ↑ | ↑ | |
| Input resistance | ↑ | – | ↑ | – | – | Aggregation in the soma impeding current flow, artificially altering capacitance. |
| Firing rate | ↑ | ↑ | – | – | – | Shift in spike initiation threshold (more hyperpolarised). |
| Rheobase | ↓ | ↓ | – | – | – | Shift in the half activation of voltage-gated sodium channels |

**Table 2 Summary of findings using recombinant truncations of tau and the identified mechanisms.**

sodium channel isoforms and therefore could affect some neurons more than others depending on the isoforms present.

## Methods

**Recombinant production of FL and tau fragments**. Constructs: FL tau (amino acids 1–441 of tau 2N4R; Uniprot ID P10636-8), as well as the amino acids 1–123 (C$^{FRAG}$) and 124–441 (N$^{FRAG}$) variants of this protein, were recombinantly produced in *E. coli*. The different fragments were PCR amplified using custom-designed primers representing the 5' and 3' sequence, respectively, with Tau cDNA (RC213312, Origene) as a template. The PCR fragment was cloned directly into pET_SUMO, a 6XHis tag and SUMO protein expression plasmid (Invitrogen), with a TA cloning site. Constructs were sequenced and transformed into *E. coli* BL21 (DE3) for expression.

SUMO fusion protein expression: *E. coli* BL21 (DE3) containing the sequence-confirmed plasmid constructs were incubated overnight in 20 ml Luria Bertani media with Kanamycin at a concentration of 50 µg/ml. The next morning, the turbid culture was used to inoculate 1 L of Luria Bertani media with Kanamycin (50 µg/ml) at +37 °C and when OD600 reached 0.5–0.7, protein expression was induced with 1.0 mM IPTG o/n at +26–28 °C. BL21 (DE3) cells overexpressing tau forms were harvested by centrifugation at 7000 rpm for 20' at +4 °C and the pellet stored at −20 °C.

Purification: The pellet was resuspended in 5 ml lysis buffer (20 mM Tris, 150 mM NaCl, 1% NP40 pH 7.5, plus a tablet of protease inhibitor cocktail from Roche) per mg of cell pellet and incubated at room temperature for 30 min. The lysate was centrifuged for 20 min at 17,000 × g, 4 °C and the supernatant was collected. The protein extract was added to Ni-NTA agarose (Novex) equilibrated with 10 mM imidazole in 1× Native buffer and incubated with rotation at +4 °C for 1 h. Ni-NTA agarose was washed with 1× Native buffer + 20 mM imidazole and 6xHis-SUMO-tau fusion protein was eluted with 250 mM imidazole in 1× Native buffer. The purified fusion protein was dialysed against 50 mM Tris, 150 mM NaCl, pH 8.0 and protein concentration was determined with BCA.

SUMO-tau fusion protein (2 mg) was cleaved by +4 °C overnight incubation in 50 µg SENP-1 SUMO protease in 50 mM Tris, 150 mM NaCl, 1 mM DTT. After cleavage, 6xHis tag and His-tagged SUMO protease were bound to Ni-NTA agarose, the flowthrough containing tag-free tau was collected and any residual bound Tau protein was eluted with imidazole gradient, starting at 20 mM. Fractions containing Tau protein were collected and dialysed against PBS. If needed, the protein was concentrated using AmiconUltra4. Polishing of the tag-free tau protein variants to remove truncates and aggregates was performed by size-exclusion chromatography[51]. We used a Superdex S200 10/300 GL column (GE Healthcare) running on an Ethan LC system (GE Healthcare), with the running buffer being 1× PBS pH 7.4 (2× PBS for tau 1–123). The following molecular weight markers from Sigma (#MWGF70-1KT) were used to estimate the elution volumes of the proteins of interest: blue dextran (2000 kDa; void volume), bovine serum albumin (66 kDa), carbonic anhydrase (29 kDa), cytochrome C (12.4 kDa), and aprotinin (6.5 kDa). Each tau variant was purified in a two-step process; 1 ml fractions from the first size-exclusion chromatography were analysed by gel chromatography followed by western blotting with specific tau antibodies to select high-protein fractions for further processing. Selected high-yield fractions were pooled and concentrated using ultrafiltration devices of appropriate molecular weight cut-offs from Amicon[51].

**Preparing of tau oligomers**. To ensure the use of equal concentrations of the tau variants, molar concentrations were estimated using molecular weights specific to each tau construct sequence. Oligomers were prepared by overnight incubation of identical concentrations of each protein construct in 1× PBS at room temperature without shaking, a method validated[22]. Monomers were treated similarly but were incubated at 4 °C overnight. The samples were immediately dry-frozen by dipping into dry ice mixed with ethanol and stored at −80 °C until use. Oligomer concentrations were expressed in equivalence of starting monomer concentrations.

**Transmission electron microscopy**. Formvar/carbon-coated 300-mesh copper grids (#S162, Agar Scientific) were glow-discharged using the ELMO system from Cordouan Technologies. Five microliters of labelled or unlabelled soluble tau aggregate preparations were pipetted onto the grid and allowed to bind for 1 min. Excess samples were removed with a strip of filter paper, and 5 µl of 2% uranyl acetate was added for 1 min. After removing the excess stain with a strip of filter paper, the grids were imaged using a JEOL-2100F transmission electron microscope.

## Electrophysiology

*Preparation of hippocampal brain slices*. All experiments were approved by the local Animals Welfare and Ethics Board (AWERB) at The University of Warwick. C57/BL6 male mice (3–4 weeks of age) were killed by cervical dislocation and decapitated in accordance with the United Kingdom Animals (Scientific Procedures) Act (1986). Parasagittal hippocampal slices (350 µM) were cut with a Microm HM 650 V microslicer in cold (2–4 °C) high Mg$^{2+}$, low Ca$^{2+}$ aCSF (artificial CSF), composed of the following: 127 mM NaCl, 1.9 mM KCl, 8 mM MgCl$_2$, 0.5 mM CaCl$_2$, 1.2 mM KH$_2$PO$_4$, 26 mM NaHCO$_3$, and 10 mM D-glucose (pH 7.4 when bubbled with 95% O2 and 5% CO$_2$, 300 mOsm). Following preparation, slices were left to recover in recording aCSF (as above but with 1 mM MgCl$_2$ and 2 mM CaCl$_2$) at 34 °C. Slices were used within 1–8 h after preparation.

*Whole-cell patch-clamp recording from pyramidal cells*. A slice was transferred to the recording chamber, submerged and perfused (2–3 ml/min$^{-1}$) with aCSF at 30 °C. Slices were visualised using IR-DIC optics with an Olympus BX151W microscope (Scientifica) and a CCD camera (Hitachi). Whole-cell current-clamp recordings were made from pyramidal cells in area CA1 of the hippocampus using patch pipettes (5–10 MΩ) manufactured from thick-walled glass (Harvard Apparatus). Pyramidal cells were identified by their position in the slice, morphology (from fluorescence imaging) and characteristics of the standard current−voltage relationship. Voltage recordings were made using an Axon Multiclamp 700B amplifier (Molecular Devices) and digitised at 20 kHz. Data acquisition and analysis were performed using pClamp 10 (Molecular Devices). Recordings from neurons that had a resting membrane potential of between –60 and –75 mV at whole-cell breakthrough were accepted for analysis. The bridge balance was monitored throughout the experiments and any recordings where it changed by >20% were discarded. Soluble tau aggregates were added to intracellular solution containing the following: 135 mM potassium gluconate, 7 mM NaCl, 10 mM HEPES, 0.5 mM EGTA, 10 mM phosphocreatine, 2 mM Mg-ATP and 0.3 mM NaGTP (293 mOsm, pH 7.2) to give a final concentration of 444 nM (20 µg/ml tau) or 133 nM (6 µg/ml tau). These concentrations were chosen to align the study with previously published findings[11]. In control experiments, vehicle (PBS) was added instead of tau. The intracellular solution was always filtered before adding tau aggregates to avoid any loss of tau in the filter.

*Stimulation protocols*. To extract the electrophysiological properties of recorded neurons, both step, ramp and more naturalistic, fluctuating currents were injected at 10-min intervals for a duration of the recordings as in refs. [11,19,20].

*Standard IV protocol*. The standard current−voltage relationship was constructed by injecting step currents from –200 pA incrementing by either 50 or 100 pA (1 s) until a regular firing pattern was induced. A plot of step current against voltage response around the resting potential was used to measure the input resistance (gradient of the fitted line).

*Rheobase ramp protocol*. To evaluate the rheobase (minimum current needed to elicit an action potential; AP) a current ramp was injected into neurons. From the baseline, a 100 ms ramp down by −50 pA, followed by a 900 ms ramp up by 150 pA, then a step back down to baseline.

*Dynamic IV protocol*. The dynamic IV curve, defined by the average transmembrane current as a function of voltage during naturalistic activity, can be used to efficiently parameterise neurons and generate reduced neural models that

accurately mimic the cellular response[20,22,25–27]. Briefly, a current waveform, designed to provoke naturalistic fluctuating voltages, was constructed using the summed numerical output of two Ornstein–Uhlenbeck processes[52] with time constants $\tau$fast = 3 ms and $\tau$slow = 10 ms. This current waveform, which mimics the background post-synaptic activity resulting from activation of AMPA and GABA$_A$ receptor channels, is injected into cells and the resulting voltage recorded (a fluctuating, naturalistic trace). The firing rate was measured from voltage traces evoked by injecting a current waveform of the same gain for all recordings (firing rate ~2–3 Hz). APs were detected by a manually set threshold and the interval between APs measured.

The dynamic transmembrane current ($I_{ion}$) can be calculated using:

$$I_{ion}(V, t) + I_{noise} = I_{inj}(t) - C\frac{dV}{dt} \qquad (5)$$

for which the injected current ($I_{inj}$) is known, the derivative (d$V$/d$t$) can be calculated from the experimentally measured voltage and the capacitance ($C$) is attained from a minimum variance procedure[26]. A scatter plot of the transmembrane current against voltage illustrates the dynamic relationship between the two, with the effects of weak background synaptic activity and other sources of high-frequency variability being accounted for as intrinsic noise ($I_{noise}$)[26]. Averaging the transmembrane current in 1 mV bins removes the time dependence of $I_{ion}$ ($V, t$) to yield the typical ionic current and a particular voltage, and thus defines the dynamic I–V curve ($I_{dyn}$):

$$I_{dyn}(V) = \text{Mean}\left[I_{ion}(V, t)\right] \qquad (6)$$

It is well-established that the exponential integrate-and-fire model[33] provides an excellent fit to the dynamic I–V curve[25–27]. The exponential integrate-and-fire (EIF) model is characterised by a voltage forcing term F(V) that is related to the $I_{dyn}$ as:

$$F(V) = \frac{-I_{dyn}(V)}{C} \qquad (7)$$

where the steady-state forcing function F(V) for the EIF model is given as:

$$F(V) = \frac{1}{\tau}\left(E - V + \triangle_{rexp}\left(\frac{V - V_T}{\triangle_T}\right)\right) = \frac{-I_{dyn}(V)}{C} \qquad (8)$$

The dynamic curve fitted to the EIF model was used to extract four parameters: membrane time constant ($\tau$), resting potential ($E$), spike-initiation threshold ($V_T$) and spike-onset sharpness ($\triangle_T$), which describes the voltage range over which an action potential linitiate[20,25–27]. Dynamic I–V curves were constructed solely from the pre-spike voltage response (subthreshold and run up to spike) with all data falling within a 200 ms window after each spike being excluded from the analysis.

**Sodium channel current recordings**. To record sodium channel currents in isolation, the intracellular and extracellular solutions used were modified from ref. [28]. The extracellular solution contained following (in mM): 124 NaCl, 25 NaHCO$_3$, 3 KCl, 1.5 CoCl$_2$, 1.0 MgSO$_4$, 0.5 NaH$_2$PO$_4$ and 30 D-glucose, equilibrated with 95% O$_2$ and 5% CO$_2$ (pH 7.4). Calcium chloride was replaced with cobalt chloride to block the voltage-gated calcium channel currents. Experiments were performed at room temperature (~22 °C to improve the quality of the voltage clamp,[28]. To reduce the amplitude of the sodium channel currents, 50 mM Na$^+$ was present in the intracellular solution: 70 Cs-gluconate, 30 Na-gluconate, 10 TEA-Cl, 5 4-AP, 10 EGTA, 1 CaCl$_2$, 10 HEPES, 4 Mg-ATP, 0.3 Na$_3$-GTP, 10 Na$_2$-phosphocreatine. Soluble full-length tau aggregates were added to the intracellular solution to give a final concentration of 444 nM (20 μg/ml tau) to align with the findings of this study and of ref. [11].

Series resistance (R$_s$) was measured throughout the recording and was typically in the range of 6–12 MΩ. Cells with R$_s$ over 15 MΩ or those that differed by more than 20% over the period of recording were discarded. R$_s$ was not compensated, but the liquid junction potential of ~8 mV was corrected. Leak currents were subtracted using the P/n protocol (1/4) and the resulting data displayed.

The stimulation protocol was adapted from ref. [28] to allow fully clamped somatic Na$^+$ channel currents to be recorded. Neurons were held at −80 mV. A depolarising pre-pulse was given (5 ms, −40 mV) to activate axonal, but not somatic, sodium channels. This was followed by a 1 ms step to −55 mV and then voltage steps (from −60 to 60 mV, 5 ms duration) were used to elicit controlled Na$^+$ channel current responses. The inter-sweep interval was 2 s. From the current responses, plots of current vs voltage or conductance vs voltage were calculated.

Conductance is given by the following equation:

$$\text{Conductance }(g) = \frac{I_{Na}}{V_{Step} - V_{Reversal}} \qquad (9)$$

The plot of conductance vs voltage was fitted with a Boltzmann equation:

$$y = \frac{A_1 - A_2}{1 + e^{(x - x_0)/dx}} + A_2 \qquad (10)$$

and then the half activation voltage and rate of activation constant were extracted and compared between control and oTau introduced neurons.

**Hodgkin–Huxley action-potential model**. The Hodgkin–Huxley (HH[29]) model is a simple but robust method for modelling the ionic conductance's that generate neuronal APs. The HH model can be applied to study voltage-gated sodium and potassium channels. It proposes that each sodium channel contains a set of three identical, rapidly responding, activation gates (the m-gates), and a single, slower-responding, inactivation gate (the h-gate)[29].

The membrane current is given by:

$$I = C_m\frac{dVm}{dt} + g_K(V_m - V_k) + g_{Na}(V_m - V_{Na}) + g_l(V_m - V_l) \qquad (11)$$

Where $I$ is the total membrane current and $C_m$ is the membrane capacitance, $g_K$, $g_{Na}$ and $g_l$ are the potassium, sodium and leak conductance's per unit area, and $V_k$, $V_{Na}$ and $V_l$ are reversal potentials.

The sodium conductance can be given by:

$$I_{Na}(t) = \bar{g}_{Na}m(V_m)^3h(V_m)(V_m - E_{Na}) \qquad (12)$$

Where $\bar{g}_{Na}$ is the maximal sodium conductance and $m$ and $h$ are quantities between 0 and 1 that are associated with sodium channel activation and inactivation, respectively. $V_m$ is the membrane potential and $E_{Na}$ is the reversal potential for sodium.

Initial parameters were assigned as follows: $\bar{g}_{Na}$ : 120 mS/cm$^2$, $\bar{g}_K$ : 36 mS/cm$^2$, $g_l$ : 0.3 mS/cm$^2$, $C_m$ : 1 μF/cm$^2$, $Vm$ : − 70 mV, $E_{Na}$ : 60 mV, $E_K$ : −88 mV, $E_l$ : −54.4 mV in line with ref. [29]. All modelling was completed using either MATLAB or Julia software platforms[53].

**Statistics and reproducibility**. We have compared each of the aggregated tau truncations with FL-oTau and with control. We have illustrated the induced changes to neuronal properties in the manuscript using normalised data (although all statistical analysis was carried out on the non-normalised data) as we are interested in the change over time (from 0 min; whole-cell breakthrough). Data were normalised to time zero when the tau constructs will not have had time to diffuse into the recorded cell, this acts as an internal control. Therefore normalising the data to the values at whole-cell breathrough makes it easier to observe any changes that occur over time and compare this between conditions.

Due to small sample sizes ($n < 15$), statistical analysis was performed using non-parametric methods on the non-normalised data: Kruskal–Wallis analysis of variance (ANOVA), Mann–Whitney and Wilcoxon signed-rank tests as required. All data are represented as mean and standard error of the mean with individual experiments represented by single data points. No post-hoc corrections have been made following these comparisons. There is considerable controversy surrounding the arbitrariness and subjectivity that can be introduced into the presentation of the results by the number of post-hoc comparisons that could or could not be undertaken[54–56]. We, therefore, prefer to present the results directly from the raw tests on a case-by-case basis. Each recorded cell is one data point. All experimental conditions were measured using multiple animals, only one cell was recorded per slice and recording conditions were interleaved to remove bias introduced from individual animals. Data points for each experimental condition were derived from a minimum of four individual animals.

A further direct comparison of all the tau constructs and control samples alongside one another is given in supplementary material. This is illustrated with both normalised and non-normalised data and a full statistical analysis of all of the conditions relative to one on the other for each parameter (Supplementary Figs. 1–6 and Supplementary Tables 1–10).

**Reporting summary**. Further information on research design is available in the Nature Research Reporting Summary linked to this article.

## Data availability

Source data underlying main figures are presented in Supplementary Data 1. All other data that support the findings of this study are available within the article and its Supplementary Material or are available from the corresponding author upon reasonable request.

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

## Acknowledgements

We would like to thank Irena Burman and Maria Olson for their contributions in helping to prepare the constructs. This work was supported by a Biotechnology and Biological Sciences Research Council-funded doctoral fellowship (E.H.). Part of this work was also supported by Alzheimer's Research UK. E.H also holds a Race Against Dementia, Barbara Naylor Charitable Trust Fellowship. TKK holds a Brightfocus fellowship (#A2020812F), and was further supported by the Swedish Alzheimer Foundation (Alzheimerfonden), the Swedish Brain Foundation (Hjärnfonden), the Swedish Dementia Foundation (Demensförbundet), the Swedish Parkinson Foundation (Parkinsonfonden), Gamla Tjänarinnor, the Aina (Ann) Wallströms and Mary-Ann Sjöbloms Foundation, the Gun and Bertil Stohnes foundation, and the Anna Lisa and Brother Björnsson's Foundation. H.Z. is a Wallenberg Scholar supported by grants from the Swedish Research Council (#2018-02532), the European Research Council (#681712), Swedish State Support for Clinical Research (#ALFGBG-720931), the Alzheimer Drug Discovery Foundation (ADDF), USA (#201809-2016862), the AD Strategic Fund and the Alzheimer's Association (#ADSF-21-831376-C, #ADSF-21-831381-C and #ADSF-21-831377-C), the Olav Thon Foundation, the Erling-Persson Family Foundation, Stiftelsen för Gamla Tjänarinnor, Hjärnfonden, Sweden (#FO2019-0228), the European Union's Horizon 2020 research and innovation programme under the Marie Skłodowska-Curie grant agreement No 860197 (MIRIADE), and the UK Dementia Research Institute at UCL. K.B. is supported by the Swedish Research Council (#2017-00915), the Alzheimer Drug Discovery Foundation (ADDF), USA (#RDAPB-201809-2016615), the Swedish Alzheimer Foundation (#AF-742881), Hjärnfonden, Sweden (#FO2017-0243), the Swedish state under the agreement between the Swedish government and the County

Councils, the ALF-agreement (#ALFGBG-715986), the European Union Joint Program for Neurodegenerative Disorders (JPND2019-466-236), and the National Institute of Health (NIH), USA, (grant #1R01AG068398-01).

## Author contributions
E.H., T.K.K. and M.J.W. designed the research; T.K.K., J.L.R., H.Z. and K.B. provided the recombinant tau samples. E.H. performed the research in the laboratory of M.J.W.; E.H. and M.J.E.R. analysed the data; E.H., T.K.K. and M.J.W. wrote the manuscript. J.L.R., H.Z., K.B. and M.J.E.R. provided edits and comments on the manuscript.

## Competing interests
The authors declare no competing interests.
