## [Transparent Peer Review File · Communications Biology]

Reviewers' comments:

Reviewer #1 (Remarks to the Author):

See attached file

Reviewer #2 (Remarks to the Author):

See attached file

Reviewer #3 (Remarks to the Author):

The authors examine here the effects of tau on the excitability of hippocampal pyramidal neurons. The main findings are: 1) N-terminal and C-terminal tau fragments have distinct effects when introduced separately in single neurons, but together act the same way as the complete tau, as tested in previous studies; and 2) tau lowers the activation voltage, steepens the activation curve, and reduces the maximal conductance of voltage-gated sodium currents, which may be responsible for at least some of the observed changes in excitability.

Although I see some problems, the study is fairly clear, the experiments are well done, the data are of good quality, the analysis is good, and the interpretation of results is more or less sound. The manuscript reads pretty well and the figures are of good quality. A definite strength is testing tau in acute experiments, by introducing the protein in single neurons via whole-cell electrophysiology, and measuring different electrophysiological properties. The effects on Nav channels are very interesting and should prompt further studies in the field.

Along with my generally positive interpretation of the manuscript, I have a few comments and suggestions that would hopefully improve the presentation.

Major comments:

I find it a bit disconcerting that the effects of different tau constructs are not compared side by side. Thus, Figs. 2 and 3 illustrate the effects of full-length (FL) and C-frag taus on neuronal excitability, Fig. 4 shows the effects of C-frag on spike initiation properties, Figs 5 and 6 document the effects of N-frag on input resistance and cellular excitability, and, finally, Fig. 7 shows the effect of FL on Nav channels.

Altogether, the authors test six (or five?) conditions: control, FL (what concentration?), C-frag, N-frag 133 nM monomers, N-frag 133 nM aggregates, and FL 444 nM. However, they don't examine the same quantities (i.e., AP shape, spike threshold, Nav activation, etc.) for each of these conditions. Presumably, some of these quantities are only affected in some experiments, but this is not clear from the figures.

Since, from the start, the authors want to test how different tau fragments affect different neuronal/channel properties, it would make sense to go through these properties one by one, comparing all six (five?) conditions in identical experiments, and then summarize the results. A few extra figures could be dedicated to explaining some of these results, such as the effect of N-frag on input resistance (Fig. 5). As it is, it's difficult to follow and understand the findings. At the very least, I would suggest a table or figure that summarizes the results (other than the graphical abstract).

I find the experiment and analysis presented in Fig. 4 unnecessarily complicated. The dynamic I-V

approach is interesting but probably only a few readers would be familiar with it, and I'm skeptical that the rest would adopt it after reading this manuscript.

The idea is to explain how the spike threshold changes. This can be very well exposed by a standard phase plot, where $CxdV/dt$ is graphed against V . Moreover, a phase plot would help the reader understand what happens with the overall depolarizing (Na^+ and Ca^{2+}) vs hyperpolarizing (K^+) voltage-gated conductances, because the positive side of the phase plot is mostly driven by Nav currents (and Cav), while the negative side is driven by Kv currents. Without doing a full pharmacological investigation, it would be useful to know how these groups of currents change. With respect to Nav channels, any change in the maximum rate of depolarization or in the inflexion point at the base of the phase plot would indicate changes in Nav kinetics or maximum conductance. I suggest showing a panel where the AP waveform is shown side by side with the phase plot.

The experiment described in Fig. 5 is difficult to interpret, and the statement in the title is not clear. The input resistance of the cell is mediated by ion channels, voltage-gated or not. Are you saying that these conductances are impeded by τ aggregation in the soma? I would be very surprised. It would mean that single channel conductances are reduced by τ , which seems doubtful to me.

On the other hand, it's conceivable that τ alters the kinetics of some channels, effectively changing their conductance at around the RMP. If some of the channels active at rest are closed by τ , then the input resistance would be higher, and a current step would produce a greater change in voltage, as shown in panel E. However, I would also expect the second pipette to record a larger voltage change, as the voltage should be the same across the soma when the electrical charges eventually equilibrate, regardless of τ .

It's possible that access/series resistance is increased by the aggregation of particles in the pipette and soma. Would this change the ionic mobility in the soma? This can be tested easily by using the R_s compensation circuitry of the patch clamp amplifier to determine the values of R_s at different times during the experiment. If the authors have not tracked and compensated for these changes in R_s , the recorded voltages (in CC) and currents (in VC) would artifactually alter their shape.

Were both pipettes filled with τ ? The cartoon should clarify that (show τ in the pipette(s) as well). In fact, it would be informative to do these experiments with τ in one pipette vs both vs none. Overall, this experiment is very interesting, but it raises the question that some of the observed changes in AP shape could be recording artifacts.

I don't know how to interpret the statement in Methods that " R_s was not compensated". It is absolutely wrong not to compensate R_s , particularly when the recorded currents are several nA. Everything will be distorted without R_s compensation. Also, in current clamp mode, R_s must be estimated and the bridge compensation must be enabled. If the authors have not taken these steps, I would have serious reason to doubt the validity (or at least the values) of all the results.

The authors find that τ reduces quite dramatically the maximum Nav conductance. Without doing a full kinetic analysis, it would still be very useful to test if Nav inactivation changes. If the inactivation curve is negatively shifted by τ , this would increase the fraction of Nav channels inactivated at the RMP and would explain the reduction in maximal conductance. A very easy experiment, and there is no reason to show activation but not inactivation, as they are effectively inseparable.

Minor comments:

Title:

I would suggest rethinking the title, if possible. As it is, it's focused on the fact that different τ

fragments have different actions. This is, of course, important but perhaps not so surprising. I find it very interesting that Nav channels (potentially) interact with tau.

Abstract:

The Abstract is not very clear about what represents previous results, and what is new to this study. For example, "Introducing N-terminally truncated tau (aa 124-441; CFRAG) removed the effects on the AP waveform and input resistance but the increase in excitability remained." I would prefix the new results with "Here, we show that...". Also, I would better motivate the experiments with Na⁺ currents, as the link is a bit weak.

"...we recorded sodium currents and found that FL-oTau lowers the activation voltage and reduced the maximal conductance, consistent with the lower spike threshold and reduction in AP amplitude." This is a bit simplistic. A negative shift in Nav activation can lower the AP threshold, but so can an increase in Nav conductance. Conversely, a reduction in Nav conductance can increase the AP threshold. Thus, the combined effects of a negative shift in Nav activation and a reduction in Nav conductance are a bit difficult to predict.

Graphical abstract:

Overall, nice and clear (but check spelling mistakes). However, it might be good to separate the physiology from the measurements, by which I mean the increase in input resistance/decrease in whole cell conductance. Are these changes only applicable to electrophysiological experiments? If tau were produced (patho)physiologically in the cell, what would be the equivalent changes, if any? Would, for example, ionic mobility in the cell decrease? Axonal conduction? Or would some membrane property change? Etc.

The last graph is a little incorrect. In the left panel, the plot of conductance vs. voltage has a slope proportional to the voltage sensitivity of the Nav activation mechanism. What is now called "rate of activation" should be called "voltage sensitivity of activation". Rate describes something that changes in time, whereas that plot (the Boltzmann) describes equilibrium properties.

The second panel should describe "a negative (or hyperpolarizing) shift in activation".

The x axes for these two plots should be labeled "Voltage", and the effects could be at the top, to match the other graphs (which, by the way, have no axes).

The inset with Na⁺ currents seems irrelevant, as there is no effect shown. Also, I'm not clear what is the role of the Nav cartoon.

What is missing from this set of figures is the effect on Nav conductance, which is dramatically reduced by tau application.

Introduction:

Page 3: comma after "previously", otherwise it reads as if oTau was previously full length and now is not.

Results:

Pg. 4, 4th paragraph: "FL oTau increased neuronal excitability, which we assumed was a result of increased Rin (Fig 2C, D, E)."

As I commented before, physiological neuronal excitability should not be equated with the response of the neuron to current injection through the recording pipette. Could you please explain this?

Fig. 2:

This figure describes changes in the AP waveform, but no AP is shown. Please add a panel with a high

resolution AP for each type of experiment (control, fl tau, fragments). It would also be useful to know if/how the AP changes in a train of spikes, and if there are long-term effects (i.e., after 40 mins). All the graphs shown are nice, but those numbers are difficult to interpret without seeing the actual changes in the waveform.

A: it would be good to know what the actual voltages are, not just a scale bar. But this information could be provided in the AP panel that I suggest to be added.

D: add axes/scale bars

F and G: Which AP in the waveform is used to calculate the height and the width? The first AP, I presume?

By "height", do you mean "peak"? If you add the AP panel, you could show what "height", etc., means.

I don't see the usefulness of showing normalized quantities when they are normalized to the same value. Moreover, it would be useful to know what the actual values are, in comparison with other studies. I highly recommend you change to absolute values.

As I already mentioned in my first comment, the figure doesn't show what the N-frag tau does. Why not? It's possible that the C and N termini must act in concert to elicit these effects. I know that a figure is shown later, but the readers would wonder, as they go through the paper.

Fig. 3.

Same comments about using absolute instead of normalized values.

I don't understand what is shown in C vs D, and I can't find clarification in the Legend (am I missing something?).

In B, I would show each trace in gray for 0 mins, and in black/blue/red for 40 mins, for each construct, respectively. In C and D, I would use black instead of gray. The overall idea is to match the color coding between B and C/D.

The caption should not be "protocol...", which is irrelevant, but something like "Tau construct changes rheobase properties in a time dependent manner".

Fig. 4.

Same comments about normalized vs. absolute.

See my second comment.

What is the difference between "threshold" and "spike onset"?

Fig. 5.

See my comment above.

Fig. 7.

What tau construct was used here? Be consistent with the names.

B vs C: it would seem that the 0 min and 20 mins data sets in C were mislabeled (red should be black and black red), because I would expect the control and tau would be identical at 0 min, and then the tau curve would diverge later on.

D and E: as before, I would much prefer to see the actual values, with a statistical test for changes over 20 min. Most electrophysiologists would be interested in knowing how many mV has the activation curve shifted. The percentage is essentially irrelevant.

At any rate, it's customary to call these quantities V_{half} and slope. DV_{half} has mV units, whereas $Dslope$ has mV or mV^{-1} units, depending on how you used the slope parameter in the Boltzmann (nominator or numerator). As I mentioned earlier, this is not rate constant, but the slope of the activation curve.

H, I: What is "mean raw conductance"? What is "mean" and what is "raw"? Raw as opposed to what? Conductance is not adimensional, should be nS.

Truncating Tau Reveals Different Pathophysiological Actions of Oligomers in Single Neurons.

We would first like to thank the reviewers and editors for their careful reading of the paper and constructive comments. We have carried out additional data analysis, amended both the text and figures and provided additional supplementary material.

General points

1. Although the data is normalised in the figures, to illustrate the changes over time, all the statistics were carried out on the raw data (which has now been added to the supplementary material)
2. To provide a narrative, the tau fragments are considered separately in the paper, rather than their effects all compared together. We have however now provided an additional comparison across all the fragments for each of the measured parameters (input resistance, membrane potential etc) in the supplementary material.
3. In previous studies, in which we have injected oligomeric species into neurons via the patch pipette, we have carried out several controls to eliminate the possibility that the observed effects are the result of blocking of the pipette tip (Kaufmann et al 2016; Hill et al 2019). We have added a summary of these controls into the discussion and have also provided measurements of the bridge balance through-out the duration of recordings.

Below we address each point in detail.

Reviewer 1

Point 1.1 Statistics: *The way that the statistics are reported is a little disingenuous. The first step is to compare the different parameters at the time point as the patch is first broken through into whole cell mode; referred to as 0 mins. This should establish that the recording conditions are similar and that any effects measured 40 minutes later are purely due to the diffusion of the Tau oligomers into the cells. The authors claim that there is no difference in any parameter at 0 mins and refer to Figure 2 and Table 1. Figure 2a and d contain some of the very few raw data in the paper in which this comparison can be observed and, at least in d the traces are rather different. Table 1 refers to “dopaminergic neurons” in the title, raising the question of whether this actually the correct data or a rather unusual typographical error (CA1 neurones being glutamatergic). Assuming the data are from the correct experiment, we then have the statistics for the comparison of the 0 mins data listed but not including the rheobase where the biggest difference occurs. Moreover, only the overall ANOVA comparison for the 3 conditions is reported without comparisons of the individual conditions (ANOVA multiple comparisons). For Rheobase for Cfrag versus control, as I can see, this would show a $adjp=0.06$ which doesn't quite make it significant but is a very strong trend. In the case of rheobase it seems that at 0 minutes the data is rather divergent but comes closer to the control level over the 40 minute period. This is also true of firing rate although the standard errors being around 20% means they would need very large sample sizes to reach statistical significance. Peak input resistance also shows a trend ($adjp=0.09$) between control and Full length Tau.*

We would like to thank the reviewer for providing this opportunity to clarify the methods we have used for statistical comparisons in this study. The technique we used is unique to our

experimental approach in that for the test groups (for example FL-oTau or C^{FARG} tau) there is an internal control at whole-cell breakthrough (when none of the tau species will have had time to diffuse into the cell). This allows an extra check to be done to ensure that the populations of neurons sampled for in each test group are not statistically different from one another at the baseline. The reviewer highlights this as a major advantage of our technique.

The reviewer points out that the traces in panel d have different baseline firing rates for each condition, this is correct and simply a result of variation between cells. When we measure the average firing rate at time 0 it is not significantly different between the groups.

In our analysis, we determine whether there are significant changes over time relative to the first recording time point (internal control, when there is no tau dialysis). We also determine whether this change over time is affected by the different preparations of tau. This process of analysis has been previously used and published by our lab (Kaufman et al, 2016, Hill et al, 2019, Hill et al, 2020) and makes complete sense given our experimental protocol.

The typographical error in table 1 has been corrected in the revised manuscript. Rheobase data has also been added into this table.

The legend to the figures has been improved in the revised manuscript to include the statistical tests performed and to confirm that there were no significant differences at time 0 (whole cell breakthrough).

In the manuscript we have analysed the data using nonparametric statistical comparisons to evaluate the significance of change over time for each condition. However, as requested by the reviewers we have included an extensive comparison of all the groups against each other over time in supplementary material (Supp Fig. 1-6, Tables 1-10). This additional analysis agrees with the findings within the manuscript.

Point 1.2 *Normalisation of data alters the results: On the basis of there being no differences at 0mins all the subsequent data are shown and analysed normalised to 0mins for the particular condition. This then completely alters some of the outcomes. Hence for the example above of Cfrag on rheobase, rather than starting differently at 0min and becoming more similar to the control over time, the data appear to start similarly and diverge over time. Similarly, with Peak input resistance for full length Tau. Consequently, as the main source of difference in some cases is at 0 minutes and the treatments tend to come together at 40 minutes, the question arises as to what the source of the difference is. Some of the data are more convincing with little difference at 0 minutes such as the action potential wave form and threshold.*

In both the original manuscript and in the revised version, all of the statistics were performed on the raw (not normalised) data. However, within the manuscript, we have presented normalised data, relative to the first time point (whole cell breakthrough), so it is easier to observe the changes over time. As mentioned above, the raw data has been added to the supplementary material.

Point 1.3 Series resistance *The authors report: "Series resistance (R_s) was measured throughout the recording and were typically in the range of 6-12 M Ω . Cells with R_s over 15 M Ω or those that differed by more than 20 % over the period of recording were discarded. R_s was not compensated, but the liquid junction potential of ~ 8 mV was corrected for." As a starting point the data for series resistance at 0, 20 and 40 minutes for each condition should be reported. It is all very well to say it was over a range of 6-12M Ω but if it was generally around 6 for control conditions and 12 for conditions when inclusions of large proteins was blocking the tip of the electrode this would explain quite a few of the results, especially where the starting data tended to be different. The authors claim that it has been previously shown that this doesn't happen, but this is very much a dynamic situation and could well vary between preparations and electrodes etc. The series resistance should be reported throughout but the fact that series resistance was not compensated for the current clamp experiments was probably not so important. Reporting series resistance and where there is large variability in the data correlating individual cell data to series resistance would be useful. However, when recording nAs of Na⁺ currents the series resistance can make a considerable difference to the results. For some reason the Na⁺ current experiments were only recorded for 20 minutes. At 0 minutes the data in F and G appear to have similar amplitudes but rather different kinetics. At no point is the raw data reported so it is not possible to tell what the effect of the normalisation will have had in this experiment. The series resistance is so intrinsic to such experiments that it definitely should be compensated for these experiments and definitely in F and G the starting and final series resistance should be reported under each condition. In summary this is potentially an interesting paper, but some very basic information is missing and the normalisation of the data distorts the outcomes. Series resistance cannot unfortunately be dismissed. It would indeed be rather surprising if inclusion of large proteins into the electrode did not have an effect on series resistance. The authors discuss the protein gathering in the soma but gathering in the electrode tip is even more likely.*

As requested, we have reported the series resistance/bridge balance at 0, 10, 20, 30 and 40 minutes for each condition (this has been added to the supplementary material sFig 8).

The reviewer makes some good points about the series resistance and the possibility that the tip of the pipette could be becoming blocked with tau. This is something we were concerned about and thus we have carried out a number of controls (as mentioned above):

- 1) We measured the bridge balance through-out recordings and observed no difference when tau oligomers were present in the patch pipette and when vehicle was present (this has been added to the supplementary material). The bridge balance was compensated for all recordings (this may not have been clear and has been added to the methods).
- 2) Previously (Hill et al 2019) we used large concentrations of another protein (BSA) in the intracellular solution and so no changes in the bridge balance.
- 3) Previously (Hill et al 2019) we made patch clamp recordings from the same pyramidal cell twice. In the first recording the electrode contained tau oligomers and in the second the electrode only contained vehicle. The changes in action potential waveform induced by tau remained during the second recording and thus cannot be a consequence of tau accumulating in the pipette.

All of these previous experiments (Hill et al 2019) were done with fluorescently labelled tau which is a significantly larger molecule than used in this paper (unlabelled tau).

We also have additional evidence that the pipette is not getting blocked during recordings:

1) We examined open tip electrodes with pressure applied and found no change in the electrode resistance with tau in the intracellular solution. Thus, the tau freely leaves the pipette tip (it has also been observed inside neurons, Hill et al 2019).

2) The major observed effect that could possibly result from an electrode tip block is the slowing of the action potential waveform and its reduction in amplitude. However, this is not a general effect of all the different oligomers, as would be expected if they just non-specifically blocked the pipette tip. Changes in the AP waveform were not observed for C^{Frag}-oTau, the large fragment but were observed for the very small fragment N^{Frag}.

3) In another set of independent experiments, we have injected much larger α -synuclein oligomers into pyramidal cells and dopaminergic neurons and we observed no slowing of the action potential waveform kinetics or change in action potential amplitude (Kaufman et al 2016; Hill et al 2021).

We have added all these points to the discussion.

We used the methods outlined in Milesco et al 2010 to allow the sodium currents to be voltage clamped. These included lowering the Na⁺ driving force, reducing the temperature, using a pre-pulse. Series resistance was monitored throughout experiments and did not significantly change (Supp mat figure 9). Furthermore, we have shown that the quality of the clamp does not change over the duration of the recordings when tau is absent Fig 7B and F. The quality of clamp may differ between recordings, but this is not of primary importance as we wish to measure changes in currents over time (as tau diffuses into the neuron).

The activation of sodium currents could be accurately fitted with a Boltzmann curve showing that they are reasonably well clamped (Supp Mat sFig 7). We have included in the supplementary data figures showing currents which were not clamped to illustrate that they are readily identifiable. The mean raw sodium channel conductance is given in panels H and I (Fig 7 manuscript)

Reviewer 2

Point 2.1 *The most important issue is the experimental design. The authors want to measure the effects of oligomeric tau fragments on electrotonic and electrogenic membrane properties. To this end, I would recommend comparing the following groups: Nfrag vs Control / Cfrag vs control / FL o-Tau vs control / Nfrag vs Cfrag vs FL o-Tau. I recommend carrying out analysis of variance on the raw data for each quantity with post-hoc t-tests with Bonferroni correction (or relevant non-parametric test if the data are not normally distributed). I recommend to clearly report the number of cells from which they have recorded and the levels of significance for each tested hypothesis (P values). These are quite routine forms of analysis. By looking at figure 2 and 6, it is not clear what the control group corresponds to in figure 2 (is it the monomer? Or just pure internal solution?) and figure 5 (is the monomer the control group?).*

We thank the reviewer for their suggestions on experimental design. In response to the points raised, we have provided graphs of comparisons for all the conditions for each parameter measured (mean and SEM) over time (Supp Matt sFig1-6 and sTables 1-10). We have provided this in both normalised and non-normalised form. We have then performed statistical

comparisons (2-way ANOVAs, Supp Matt) for all the groups against each other as requested and demonstrated that these results agree with the findings in the manuscript.

We have not used corrections for multiple comparisons as requested by the reviewer as we don't feel that they are the best way to analyse the data. There is considerable controversy as to whether multiple-comparison corrections should be applied see for example:

No adjustments are needed for multiple comparisons. Rothman KJ, Epidemiology (1990) 1:43-46:

Multiple comparison procedures: the practical solution. Saville DJ, The American Statistician (1990) 44:174-180.

Rules and judgements in statistics: three examples. Stewart-Oaten A. Ecology (1995) 76:2001-2009.

These papers highlight the arbitrariness and subjectivity that would be introduced into the presentation of the results by the number of post-hoc comparisons we could have chosen (or chosen not to do). We therefore prefer to present the results directly from the raw tests on a case-by-case basis and have made it explicit in the methods that no post-hoc corrections have been made.

To clarify the species used, control in figure 2 and 5 refers to internal patch solution with no added tau species, just vehicle. In figure 6 (the N^{FRAG} comparison), there is no 'control' data shown however the full-length oTau results are shown as dotted lines (replicated data from earlier in the paper) as a means of comparison with the other two conditions.

Point 2.2 *In figure 2 the authors claim that Nfrag aggregates look different from the Cfrag and FL-oTau ones, when observed with a TEM. It is not clear to me how this was measured. By just looking at the pictures, this difference is not obvious to me.*

We have clarified this in the text. In the figure we intended to illustrate that all the fragments can form aggregates. We have not done any further characterisation of the structure of the aggregates. This is something we hope to investigate in further studies.

Point 2.3 *It is not clear why the authors show normalized data. The plasma membrane excitability properties are usually normally distributed within a homogeneous population of neurons. The plots do not show the real values of the properties (the reader needs to consult the table for that). The effect of time on certain properties is likely due to the time required by the oligomers to diffuse from the pipette into the cell. I recommend measuring when the value of each property reaches a steady state and compare only that point between groups. In case the effect of time and treatment wants to be investigated within the same envelope, I recommend to still carry out this analysis on the raw values. Finally, it would be useful to show the data as scatter-box plots, as these are the ones that show both the descriptors of the population (average +/- SEM) and the individual data-points.*

As mentioned above, the figures show normalised data but all the statistics were carried out on the raw data values. As mentioned above, normalised data is plotted in the figures to make it for easier for the reader to observe the changes over time. However, to respond fully to this point, non-normalised data is now presented in the Supp Mat (Supp Fig. 1-6, Tables 1-10).

The reviewer also makes the point that the time for tau to have an effect may vary between cells (depending on diffusion) and therefore it is more interesting to pick a late time point (e.g., 40 mins) to compare to the control at 0 minutes. This is what we have done in the manuscript but perhaps this was not clear. We have therefore improved the results text in the revised version to improve clarity.

Point 2.4 *The voltage-clamp recordings have been carried out on native neurons, in brain slices. The authors claim they can effectively space clamp the neuronal soma by inactivating the axonal voltage-gated sodium channels (mostly Nav1.6) with a brief depolarizing current step preceding the IV curve. I do not think this method is sufficient to isolate somatic currents. In fact, Nav1.2 and other slower channels are expressed also in other processes of the neuron, including the soma. In addition, the complex geometry of the neuron does not allow to use the measure of capacitance as an indirect measure of the plasma membrane surface area, to calculate current and conductance densities. Finally, all the voltage clamp data are presented as conductances, calculated from the cell currents. To assess the effects of treatment on the voltage-gated currents and avoid biases coming from potential effects on the cell size and membrane area, the Boltzmann fits should be carried out on specific conductances, calculated by dividing each conductance by the capacitance of the cell. This can be done only if the cell has a regular shape, such as a spheroid. For these reasons, I recommend carrying out these recordings again by using outside-out nucleated macropatches or at least re-analyse the data showing specific conductances and discussing the limits of the space clamping in a complex neuronal structure. In the discussion, the authors mention why they decided to not use macropatches, as they have been defined “less physiologically relevant because the distribution, density and expression pattern of channel will be different from what is found in situ”. It is not clear to me what the authors meant by this.*

We have taken the approach outlined in the paper published in the Journal of Neuroscience from the lab of Bruce Bean (Milescu et al 2010). This paper outlined a number of procedures that allow sodium currents to be voltage clamped in hippocampal pyramidal cells in slices: recordings are carried out at room temperature, there is a lowering of Na⁺ driving force and a pre-pulse to inactivate some of the channels. They showed (Milescu et al 2010) that the kinetics of the currents recorded in slices using these methods were not significantly different from that observe in dissociated neurons, where normally the quality of the voltage clamp is much higher.

In supplementary material we have provided examples of sodium currents (Supp material sFig 7) which are not effectively clamped to show that recordings of unclamped sodium currents can be readily identified and excluded from analysis.

We have also demonstrated that the ability to fit the current activation with a Boltzmann curve and parameterise is not affected by plotting the raw current, conductance or normalised conductance. The results from all 3 approaches agree with the findings in the manuscript.

The use of outside-out nucleated patches is a very good idea and will probably provide better voltage clamp control of the sodium currents than the methods that we have used. However, there is one potential problem, the volume of the patch will be much smaller than a whole pyramidal cell. Thus, the concentration of tau that reaches the channels may be much higher

and thus have a greater effect. It would be difficult to accurately titrate the concentration of tau, so it is comparable to experiments done in whole cells. We have made this point in the discussion.

Point 2.5 *It is not clear what form of tau they use in the voltage clamp experiments: Cfrag, Nfrag, FL-otau? The authors should carry out these recordings with each one of these treatments if they want to support the hypotheses arising from the observations collected with the current clamp experiments.*

We thank the reviewer for the opportunity to clarify this. The tau used in these voltage clamp experiments was the full-length tau, as we wished to evaluate whether both of the observed effects (on action potential waveform and excitability) could be a result of an interaction with NaV channels. While it would be interesting to repeat these experiments with different tau truncations to find the potential interaction sites, this will form the basis of future studies and lies outside of the scope of this current investigation.

Point 2.6 *It is not clear how the AP threshold was calculated. A common definition of threshold voltage is: the voltage at which the dV/dT exceeds the value of 10V/s. In fact, it has been shown that at that rate, more than 50% of the Nav channels are activated <https://www.ncbi.nlm.nih.gov/pmc/articles/PMC2900290/>. The methods section explaining the way you have calculated the AP thr is very lengthy, but it does not provide this key information.*

As described in the manuscript we used the established dynamic IV protocol to parameterise the neurons. The original publication provides a full description of the underlying mathematical modelling (Badel et al, 2008). To address the point raised by the reviewer we have also carried out the additional analysis of AP threshold suggested by the reviewer for the cell presented in Figure 4A. When threshold is instead defined as point where dV/dT exceeds the value of 10V/s, CFrag oTau mediates a reduction in AP threshold of 6 mV, consistent with our results from the dynamic IV method and reflecting the increase in excitability.

Point 2.7 *In figure 5 the authors show the effects of Nfrag at 444 nM on the pyramid input resistance and potentially ascribe it to the accumulation of insoluble tau inside the cell. From the look of the green traces, where the action potentials almost disappear, it seems that the access resistance of the pipette increased. Can you report the access resistance values you have recorded with the bridge balance function? I would expect a significance increase of that value.*

We have added the bridge balance measurements for these experiments to the supp matt (sFig. 8) There was no change in the bridge balance over the period of recording. This is consistent with our previous studies (Hill et al, 2019; Kaufmann et al 2016). As mentioned above (point 1.3) we have carried out several controls to exclude the possibility of tau increasing pipette resistance.

Reviewer 3

Point 3.1 *I find it a bit disconcerting that the effects of different tau constructs are not compared side by side. Thus, Figs. 2 and 3 illustrate the effects of full-length (FL) and C-frag taus on neuronal excitability, Fig. 4 shows the effects of C-frag on spike initiation properties, Figs 5*

and 6 document the effects of N-frag on input resistance and cellular excitability, and, finally, Fig. 7 shows the effect of FL on Nav channels. Altogether, the authors test six (or five?) conditions: control, FL (what concentration?), C-frag, N-frag 133 nM monomers, N-frag 133 nM aggregates, and FL 444 nM. However, they don't examine the same quantities (i.e., AP shape, spike threshold, Nav activation, etc.) for each of these conditions. Presumably, some of these quantities are only affected in some experiments, but this is not clear from the figures. Since, from the start, the authors want to test how different tau fragments affect different neuronal/channel properties, it would make sense to go through these properties one by one, comparing all six (five?) conditions in identical experiments, and then summarize the results. A few extra figures could be dedicated to explaining some of these results, such as the effect of N-frag on input resistance (Fig. 5). As it is, it's difficult to follow and understand the findings. At the very least, I would suggest a table or figure that summarizes the results (other than the graphical abstract).

There is a table in the discussion comparing the effects of the fragments side by side for each comparison, highlighting the varied effects for each of the conditions. We have also added comparisons for all of the conditions side by side for each of the measured parameters into the supplementary material (sFigs 1 to 6 and sTables 1-10) to address this point (see above).

Point 3.2 *I find the experiment and analysis presented in Fig. 4 unnecessarily complicated. The dynamic I-V approach is interesting but probably only a few readers would be familiar with it, and I'm skeptical that the rest would adopt it after reading this manuscript. The idea is to explain how the spike threshold changes. This can be very well exposed by a standard phase plot, where $Cx dV/dt$ is graphed against V . Moreover, a phase plot would help the reader understand what happens with the overall depolarizing (Na^+ and Ca^{2+}) vs hyperpolarizing (K^+) voltage-gated conductances, because the positive side of the phase plot is mostly driven by Nav currents (and Cav), while the negative side is driven by Kv currents. Without doing a full pharmacological investigation, it would be useful to know how these groups of currents change. With respect to Nav channels, any change in the maximum rate of depolarization or in the inflexion point at the base of the phase plot would indicate changes in Nav kinetics or maximum conductance. I suggest showing a panel where the AP waveform is shown side by side with the phase plot.*

We would respectfully disagree that the method is unnecessarily complicated. It is the appropriate method for extracting the ionic current under these circumstances and is also well established, with the original paper cited over 200 times (Badel et al 2008). Confirmation that it is the right approach can be seen from the current balance equation

$$CdVdt + I_{ion} = I_{injected}.$$

where it is clear that the rate of change of voltage (CdV/dt) does not isolate the ionic current but rather the difference between the ionic current and $I_{injected}$ and so is not the quantity required under conditions of current stimulation.

"Point 3.3 *The experiment described in Fig. 5 is difficult to interpret, and the statement in the title is not clear. The input resistance of the cell is mediated by ion channels, voltage-gated or not. Are you saying that these conductances are impeded by tau aggregation in the soma? I*

would be very surprised. It would mean that single channel conductances are reduced by tau, which seems doubtful to me. On the other hand, it's conceivable that tau alters the kinetics of some channels, effectively changing their conductance at around the RMP. If some of the channels active at rest are closed by tau, then the input resistance would be higher, and a current step would produce a greater change in voltage, as shown in panel E. However, I would also expect the second pipette to record a larger voltage change, as the voltage should be the same across the soma when the electrical charges eventually equilibrate, regardless of tau. It's possible that access/series resistance is increased by the aggregation of particles in the pipette and soma. Would this change the ionic mobility in the soma? This can be tested easily by using the R_s compensation circuitry of the patch clamp amplifier to determine the values of R_s at different times during the experiment. If the authors have not tracked and compensated for these changes in R_s , the recorded voltages (in CC) and currents (in VC) would artifactually alter their shape. Were both pipettes filled with tau? The cartoon should clarify that (show tau in the pipette(s) as well). In fact, it would be informative to do these experiments with tau in one pipette vs both vs none. Overall, this experiment is very interesting, but it raises the question that some of the observed changes in AP shape could be recording artifacts.

We have amended the text in the revised manuscript to make it clearer to the reader:

“To test the possibility that this rapid change to whole-cell resistance and capacitance could be due to the aggregation of the N^{FRAG} tau in the soma impeding current flow, we performed a subset of experiments where two simultaneous whole-cell patch clamp recordings were made from the soma of the same CA1 pyramidal neuron (Fig 5B; see methods). We injected current into one pipette and then measured the ratio of the voltage responses in the “inject” pipette and in the “recording” pipette. We reasoned that if the N^{FRAG} tau increases whole cell resistance then less current will leak out from the neuron and thus the voltage response measured from both pipettes will be increased without a change in ratio. If however the N^{FRAG} aggregates in the soma and interferes with current flow between the pipettes then the ratio of the voltage responses (recording/inject) will reduce as less current will reach the “record” pipette.

We injected a 200 pA (1s) hyperpolarising step via the “inject” pipette and measured voltage responses in both the inject and record pipettes (Fig 5C). As the current needs to travel to the ‘record’ pipette, the voltage response was always slightly smaller than the response measured by the ‘inject’ pipette. In cells where both pipettes were filled with vehicle, there was no change in the relative amplitude of the voltage responses between 0 mins (whole-cell breakthrough) and 5 mins (Fig 5D). However, when N^{FRAG} tau was introduced via the ‘inject’ pipette, there was a large increase in the voltage response measured via the inject pipette indicative of an increase in cell resistance. However, this was not reflected by an increase in the amplitude of the voltage step as measured by the “record” pipette (Fig 5E, F). Thus, the ratio of record/inject was significantly reduced (Fig 5E, F). This result confirms that at this concentration, N^{FRAG} oTau is accumulating in the soma, impeding the flow of current between the two pipettes, resulting in an ‘increase in the apparent input resistance and a decrease in the observed capacitance due to the cell appearing electronically smaller”.

We did not observe any changes in the bridge balance during these recordings (added to Supp Matt sFig. 8) although there were large changes in input resistance and spike amplitude. This has been added to the text.

Point 3.4 *I don't know how to interpret the statement in Methods that "Rs was not compensated". It is absolutely wrong not to compensate Rs, particularly when the recorded currents are several nA. Everything will be distorted without Rs compensation. Also, in current clamp mode, Rs must be estimated and the bridge compensation must be enabled. If the authors have not taken these steps, I would have serious reason to doubt the validity (or at least the values) of all the results.*

We would like to thank the reviewers for the opportunity to clarify this misunderstanding which has most likely occurred due to a lack of clarity in the methods text for which we apologise. To confirm, the bridge balance was compensated for all current clamp recordings (this has been made clear in the methods). A graph of the bridge balance over time for all conditions is now provided in Supp Mat (sFig 8). For the voltage clamp recordings please see the comments above.

Point 3.5 *The authors find that tau reduces quite dramatically the maximum Nav conductance. Without doing a full kinetic analysis, it would still be very useful to test if Nav inactivation changes. If the inactivation curve is negatively shifted by tau, this would increase the fraction of Nav channels inactivated at the RMP and would explain the reduction in maximal conductance. A very easy experiment, and there is no reason to show activation but not inactivation, as they are effectively inseparable.*

While we agree that this would be an interesting thing to test in a future study it was outside the scope of this one. We have added this possibility to the discussion.

Point 3.6 *I would suggest rethinking the title, if possible. As it is, it's focused on the fact that different tau fragments have different actions. This is, of course, important but perhaps not so surprising. I find it very interesting that Nav channels (potentially) interact with tau.*

As requested, we have changed the title to ensure that it frames the effects on Na Channels:

'Truncating Tau Reveals Different Pathophysiological Actions of Oligomers in Single Neurons and Reveals Sodium Channels as a Novel Target'

Point 3.7 *The Abstract is not very clear about what represents previous results, and what is new to this study. For example, "Introducing N-terminally truncated tau (aa 124-441; CFRAG) removed the effects on the AP waveform and input resistance but the increase in excitability remained." I would prefix the new results with "Here, we show that...". Also, I would better motivate the experiments with Na⁺ currents, as the link is a bit weak.*

We have re-written this for clarification:

Tau protein is involved in maintaining the structural integrity of neurons. In tauopathies, including Alzheimer's disease, tau forms oligomers, which modulate neuronal function. We have previously shown that the introduction of oligomeric full-length human tau (aa 1-441; FL-oTau) into pyramidal neurons decreases whole-cell conductance, increases excitability and

changes the action potential (AP) waveform. Here we show that introducing N-terminally truncated tau (aa 124-441; C^{FRAG}) abolished the effects on the AP waveform and input resistance (IR) but the increase in excitability remained. A hyperpolarising shift in spike threshold underlies this increase. The N-terminal fragment (aa 1-123; N^{FRAG}) increased IR and changed the AP waveform. Thus, the two truncations can recapitulate the effects of FL-oTau. To investigate the underlying mechanisms that change the action potential waveform, we recorded sodium currents and found that FL-oTau lowers activation voltage and reduces maximal conductance, consistent with the lower spike threshold and reduction in AP amplitude.

Point 3.8 *"...we recorded sodium currents and found that FL-oTau lowers the activation voltage and reduced the maximal conductance, consistent with the lower spike threshold and reduction in AP amplitude." This is a bit simplistic. A negative shift in Nav activation can lower the AP threshold, but so can an increase in Nav conductance. Conversely, a reduction in Nav conductance can increase the AP threshold. Thus, the combined effects of a negative shift in Nav activation and a reduction in Nav conductance are a bit difficult to predict.*

Yes, it is possible that increase in Nav conductance could lower AP threshold but this not the case here as we see a fall in NaV conductance.

Point 3.9 *Graphical abstract: Overall, nice, and clear (but check spelling mistakes). However, it might be good to separate the physiology from the measurements, by which I mean the increase in input resistance/ decrease in whole cell conductance. Are these changes only applicable to electrophysiological experiments? If tau were produced (patho) physiologically in the cell, what would be the equivalent changes, if any? Would, for example, ionic mobility in the cell decrease? Axonal conduction? Or would some membrane property change? Etc. The last graph is a little incorrect. In the left panel, the plot of conductance vs. voltage has a slope proportional to the voltage sensitivity of the Nav activation mechanism. What is now called "rate of activation" should be called "voltage sensitivity of activation". Rate describes something that changes in time, whereas that plot (the Boltzmann) describes equilibrium properties. The second panel should describe "a negative (or hyperpolarizing) shift in activation". The x axes for these two plots should be labeled "Voltage", and the effects could be at the top, to match the other graphs (which, by the way, have no axes). The inset with Na⁺ currents seems irrelevant, as there is no effect shown. Also, I'm not clear what is the role of the Nav cartoon. What is missing from this set of figures is the effect on Nav conductance, which is dramatically reduced by tau application.*

We have checked and revised the graphical abstract as per the requests of the reviewer.

Point 3.10 *Introduction: Page 3: comma after "previously", otherwise it reads as if oTau was previously full length and now is not.*

We have added the comma as requested.

Point 3.11 *Results: Pg. 4, 4th paragraph: "FL τ increased neuronal excitability, which we assumed was a result of increased R_{in} (Fig 2C, D, E)." As I commented before, physiological neuronal excitability should not be equated with the response of the neuron to current injection through the recording pipette. Could you please explain this?*

It is clear that neurons will not physiologically experience the sustained depolarisation that is used during a standard step current-voltage relationship. However, we have taken the additional approach of injecting a naturalistic current that mimics the activation of AMPA and GABAA receptors (Fig 2 D). We have taken this approach in previous publications (Hill et al 2019; Hill et al 2021; Kaufmann et 2016). We have changed the text to say, "a correlate of excitability".

Point 3.12 *Fig. 2: This figure describes changes in the AP waveform, but no AP is shown. Please add a panel with a high-resolution AP for each type of experiment (control, fl tau, fragments). It would also be useful to know if/how the AP changes in a train of spikes, and if there are long-term effects (i.e., after 40 mins). All the graphs shown are nice, but those numbers are difficult to interpret without seeing the actual changes in the waveform. A: it would be good to know what the actual voltages are, not just a scale bar. But this information could be provided in the AP panel that I suggest being added. D: add axes/scale bars F and G: Which AP in the waveform is used to calculate the height and the width? The first AP, I presume? By "height", do you mean "peak"? If you add the AP panel, you could show what "height", etc., means. I don't see the usefulness of showing normalized quantities when they are normalized to the same value. Moreover, it would be useful to know what the actual values are, in comparison with other studies. I highly recommend you change to absolute values. As I already mentioned in my first comment, the figure doesn't show what the N-frag tau does. Why not? It's possible that the C and N termini must act in concert to elicit these effects. I know that a figure is shown later, but the readers would wonder, as they go through the paper.*

We have added a high-resolution image of the APs for the 3 conditions as requested in Figure 2. Axes and scale bars have been added to panels F and G. We haven't measured effects after 40 minutes as this was outside the scope of the study.

The AP waveform measurements (peak/height and width) are calculated as an average of all of the action potentials generated in response to the injection of the noisy current. This is a much more realistic approach than the standard approach of measuring from the step responses in terms of physiology. Height refers to the amplitude (to the peak of the action potential). We have clarified this in the manuscript.

While the reviewer is correct the N^{FRAG} data is not found in this figure, it is shown later in the paper to allow the narrative to flow better. The idea of the truncation strategy is to show that these fragments can have distinct effects from each other. To address this concern of the reviewer, we have added plots of the raw data (Supp Mat) of all of the conditions compared alongside each other including N^{FRAG} .

Point 3.13 *Fig. 3. Same comments about using absolute instead of normalized values. I don't understand what is shown in C vs D, and I can't find clarification in the Legend (am I missing*

something?). In B, I would show each trace in grey for 0 mins, and in black/blue/red for 40 mins, for each construct, respectively. In C and D, I would use black instead of gray. The overall idea is to match the colour coding between B and C/D. The caption should not be "protocol...", which is irrelevant, but something like "Tau construct changes rheobase properties in a time dependent manner".

We have modified the legend to figure 3 as requested to clarify. We have addressed the concerns over normalised data (see above). We have modified the captions to reflect the figure better.

Point 3.14 *Fig. 4. Same comments about normalized vs. absolute. See my second comment. What is the difference between "threshold" and "spike onset"?*

We have added plots of the raw data to the supplementary material (see above).

To clarify, threshold refers to the value that needs to be surpassed in order for an action potential to be fired. Spike onset refers to how quickly the action potential occurs following the threshold. This has been made clearer in the text of the revised manuscript.

Point 3.15 *Fig. 5. See my comment above.*

We have added plots of the raw data (Supp Mat, see above)

Point 3.16 *Fig. 7. What tau construct was used here? Be consistent with the names. B vs C: it would seem that the 0 min and 20 mins data sets in C were mislabeled (red should be black and black red), because I would expect the control and tau would be identical at 0 min, and then the tau curve would diverge later on. D and E: as before, I would much prefer to see the actual values, with a statistical test for changes over 20 min. Most electrophysiologists would be interested in knowing how many mV the activation curve has shifted. The percentage is essentially irrelevant. At any rate, it's customary to call these quantities V_{half} and slope. DV_{half} has mV units, whereas $Dslope$ has mV or mV^{-1} units, depending on how you used the slope parameter in the Boltzmann (nominator or numerator). As I mentioned earlier, this is not rate constant, but the slope of the activation curve. H, I: What is "mean raw conductance"? What is "mean" and what is "raw"? Raw as opposed to what? Conductance is not adimensional, should be nS.*

The tau construct used in these experiments was the full-length oligomers.

The data is correctly labelled in the figure. The curves are normalised to illustrate the shift over-time. The non-normalised superimposed conductances are plotted in Supp Matt (SFig 7 panel G).

The mV that the activation curve has shifted is supplied as a shift in V_{half} (in mV).

Mean raw conductance is the mean value of the conductances. The term "raw" means it has been normalised to the conductance at time zero.

We have added units to panel H (nS)

Reviewers' comments:

Reviewer #2 (Remarks to the Author):

Truncating Tau Reveals Different Pathophysiological Actions of Oligomers in Single Neurons
Response to the rebuttal

The overall quality of the manuscript has significantly improved. However, there are still some outstanding issues. Below I respond to the authors answers to my comments, following the original order.

Major concerns

1) This comment has been largely addressed. However, there are certain issues standing: a) I still could not find the number of replicates (single cells?) for each experimental group and b) multiple comparisons correction should not be used for screening large bodies of data, true, but it is undoubtedly an essential tool for reducing the risk of type 1 errors when testing a specific hypothesis. That is, the statistical envelope should be chosen before obtaining the data and not the other way around. If multiple comparisons will not be used, I then suggest to only test the effect of the treatment over each property. I see that other reviewers share my concerns over the statistical treatment of these data. Therefore, I strongly suggest considering this advice to avoid to incur into the risk of type 1 errors. Alternatively, please clarify in the methods section the reason why you have decided to not carry out multiple comparisons and how you trust that the differences observed are not the consequence of a type 1 error.

2) All good here.

3) All good here as the non-normalized data are now presented as well. I still do not see the rationale for normalizing these data. This procedure removes degrees of freedom to the statistical envelope and does not aid to address the question at hand. Once again, other reviewers share my doubts. Please, consider removing the normalised data or to provide a more convincing justification for this methodological choice.

4) For the use of somatic outside out nucleated patches, provided that the patch remains stable for few minutes to allow the diffusion of the fragment across the whole cell, there should not be significant differences in concentration. Also, the signals picked up in whole-cell configuration are mostly relying on components originating in the soma, while being biased by space-clamping problems due to the complex geometry of the neuron. I still do not see how your approach can provide an accurate estimate of the current densities, as the references you have provided do not provide answer to this question. I recommend to better clarify in the discussion the pros and cons of your approach.

5) This should still be better clarified in the text and in the figure legends.

6) All good here.

7) All good here.

8) The authors use different concentrations of the tau fragments: 444 nM and 133 nM. How and why have these specific values been chosen? This has not been addressed.

Minor concerns.

None of these have been addressed.

1) In the discussion you underlie how the Cfrag leads to increased excitability via the hyperpolarization of the AP threshold. It has been previously observed in Tau35 mice, expressing a 35 KDa C-term fragment of tau, a similar hyperpolarization of the AP threshold, but also a voltage dependent decrease of the input resistance

(<https://www.ncbi.nlm.nih.gov/pmc/articles/PMC5654728/>). You should cite this paper and discuss the similarities and differences with your observations.

2) In the introduction, "liming step" should be "limiting step".

Reviewer #3 (Remarks to the Author):

This is a nice manuscript, and I'm satisfied that the authors have addressed all my comments and

suggestions.

I think the choice of using whole-cell recordings with the voltage prepulse technique is sound, and the recorded Nav currents show satisfactory clamp. Nucleated patches can produce perfect clamp, but are noisy and cause changes in Nav kinetic properties.

I have a few very minor comments:

Pg. 8: "It is feasible that some of the changes"  "conceivable".

"which is the opposite to what be expected"  "what would be"

Fig. 7J - "Simulated"

Response to reviewers

We would first like to thank the reviewers and editors for their careful reading of the revised version of our manuscript and providing further constructive comments which will improve the finished version. We amended the text and figures to clarify all remaining queries. Responses to specific comments are outlined below.

Editor

Please check the axes in the supplementary material. Some lack the units (e.g. APH in mV, IR MOhm etc.).

All axes have been checked and now have the required units.

Reviewer #2 (Remarks to the Author):

1a) I still could not find the number of replicates (single cells?) for each experimental group

We would thank the reviewer for highlighting this omission. We have now ensured all n numbers (both of cells and animals, means, SEMs and P values) are given in the figure legends.

1b) Please clarify in the methods section the reason why you have decided to not carry out multiple comparisons and how you trust that the differences observed are not the consequence of a type 1 error.

We have made it clear, in the statistical analysis section of the manuscript, that no corrections were made for the multiple comparisons. We have added an explanation for why we have taken this approach with supporting references.

In the supplementary material, in the previous revision, we provided a side-by-side comparison of all of the conditions for each of the parameters. We analysed this data using 2-Way ANOVAs with multiple comparisons and confirmed that the results were in fact no different to the findings in the manuscript. We think this is compelling evidence that the data is robust and reliable.

3) Please, consider removing the normalised data or to provide a more convincing justification for this methodological choice.

We have fully clarified the reasons for displaying normalised data in the methods section of the manuscript (statistical analysis). Our experimental method provides an internal control for each recording, as at whole cell breakthrough the tau constructs will not have diffused into the neuron. We are interested in a change in the neuronal parameters over time starting from this time point as the tau constructs diffuse into the neuron to have their effects. This method reduces the effect of variation in parameters across recordings and it is much easier to observe the changes across time.

To respond to the reviewers concerns we have previously provided the full non-normalised (raw) dataset in the supplementary data. We have confirmed by providing all the raw data and completing statistical analysis of all conditions side by side that there the no difference in the conclusions drawn from the normalised and the raw sets of data. We have fully justified this in the manuscript text.

As mentioned before although the normalised data is displayed in the manuscript all statistics were carried out on the raw data.

4) For the use of somatic outside out nucleated patches, provided that the patch remains stable for few minutes to allow the diffusion of the fragment across the whole cell, there should not be significant differences in concentration. Also, the signals picked up in whole-cell configuration are mostly relying on components originating in the soma, while being biased by space-clamping problems due to the complex geometry of the neuron. I still do not see how your approach can provide an accurate estimate of the current densities, as the references you have provided to not provide answer to this question. I recommend to better clarify in the discussion the pros and cons of your approach.

While the reviewer makes good points about the use of somatic outside out nucleated patches, we strongly believe that the approach we have taken is the best way to answer our question. As reviewer three highlights below:

'I think the choice of using whole-cell recordings with the voltage pre-pulse technique is sound, and the recorded Nav currents show satisfactory clamp. Nucleated patches can produce perfect clamp but are noisy and cause changes in Nav kinetic properties.' – Reviewer 3

The protocol we used for recording Na channel currents is published and is well cited. We have provided additional evidence (in the previous revision) that the currents are well clamped and not subject to space clamp problems. We have fully explained the potential issues with space clamp, the different methods that can be used and the advantages of our method over other approaches in the discussion on page 10.

5) This should still be better clarified in the text and in the figure legends.

We have clarified that these experiments were FL-oTau (444 nM) vs control in the results, the methods and the figure legend and have explained why the full-length version of tau was used.

8) The authors use different concentrations of the tau fragments: 444 nM and 133 nM. How and why have these specific values been chosen? This has not been addressed.

This work extends a previously published study which highlighted the changes to neuronal properties with full length tau. In this study we have investigated what underlies these changes. Therefore, the concentrations used in this study were kept the same as used in the previous publication for consistency. We have added this information to both to the introduction and the methods section.

Minor concerns.

1) In the discussion you underlie how the Cfrag leads to increased excitability via the hyperpolarization of the AP threshold. It has been previously observed in Tau35 mice, expressing a 35 KDa C-term fragment of tau, a similar hyperpolarization of the AP threshold, but also a voltage dependent decrease of the input resistance (<https://www.ncbi.nlm.nih.gov/pmc/articles/PMC5654728/>). You should cite this paper and discuss the similarities and differences with your observations.

We have added this reference into our discussion to highlight that a hyperpolarisation of threshold has been reported in a tau model previously. There are considerable differences between the approaches used in this paper and in our studies. For example they use a transgenic model, the cells are held at set voltages rather than recorded at their resting membrane potentials etc. This makes it difficult to directly compare the observations

2) In the introduction, "liming step" should be "limiting step". Has been corrected as requested

Reviewer #3 (Remarks to the Author):

This is a nice manuscript, and I'm satisfied that the authors have addressed all my comments and suggestions.

I think the choice of using whole-cell recordings with the voltage pre-pulse technique is sound, and the recorded Nav currents show satisfactory clamp. Nucleated patches can produce perfect clamp but are noisy and cause changes in Nav kinetic properties.

We thank the reviewer for their support of the technique. We feel that the supplementary data provides strong evidence that the currents are well clamped, and that this technique is well suited to answer the questions that we are asking.

I have a few very minor comments:

Pg. 8: "It is feasible that some of the changes"  "conceivable". Changed as requested

"which is the opposite to what be expected"  "what would be" Changed as requested

Fig. 7J - "Simulated" Changed as requested

REVIEWERS' COMMENTS:

Reviewer #2 (Remarks to the Author):

I am satisfied with the rebuttal and the modifications that the authors provided.

One last minor observation is that in the text the input resistance is abbreviated as "Rin", whereas in the plots it is referred to as IR. I recommend to make sure that the acronyms are consistent throughout the manuscript, to avoid confusion.